# Synthesis of bioactive (1→6)-β-glucose branched poly-amido-saccharides that stimulate and induce M1 polarization in macrophages

Ruiqing Xiao[1,2,3,7], Jialiu Zeng [4,5,7], Eric M. Bressler[4], Wei Lu[6] & Mark W. Grinstaff [1,4] ✉

β-Glucans are of significant interest due to their potent antitumor and immunomodulatory activities. Nevertheless, the difficulty in purification, structural heterogenicity, and limited solubility impede the development of structure-property relationships and translation to therapeutic applications. Here, we report the synthesis of a new class of (1→6)-β-glucose-branched poly-amido-saccharides (PASs) as β-glucan mimetics by ring-opening polymerization of a gentiobiose-based disaccharide β-lactam and its copolymerization with a glucose-based β-lactam, followed by post-polymerization deprotection. The molecular weight ($M_n$) and frequency of branching (FB) of PASs is readily tuned by adjusting monomer-to-initiator ratio and mole fraction of gentiobiose-lactam in copolymerization. Branched PASs stimulate mouse macrophages, and enhance production of pro-inflammatory cytokines in a FB-, dose-, and $M_n$-dependent manner. The stimulation proceeds via the activation of NF-κB/AP-1 pathway in a Dectin-1-dependent manner, similar to natural β-glucans. The lead PAS significantly polarizes primary human macrophages towards M1 phenotype compared to other β-glucans such as lentinan, laminarin, and curdlan.

The search for bioactive polysaccharides is an active research area in food science, cosmetics, and medicine[1,2] Mushroom-derived β-glucans have been extensively studied in the last half century because they exhibit strong antitumor and immunomodulatory activities with minimal side effects[3,4]. These polysaccharides are not directly cyto-toxic to cancer cells but instead exert tumoricidal effects via activation of the host immune system[5]. For example, schizophyllan and lentinan (Fig. 1), are approved in Japan for clinical use in cancer treatment[6]. Recent clinical studies show that, compared to chemotherapy alone, chemo-immunotherapy with lentinan prolongs patient survival with advanced gastric cancers[7,8]. β-Glucans typically consist of a linear and rigid (1→3)-β-linked D-glucopyranose backbone with (1→6)-β-linked glucopyranose branches occurring at different frequencies[9]. In

aqueous solution, β-glucans usually form triple helical conformations[10]. Accumulating evidence demonstrates that the biological activities of β-glucans are influenced by their frequency of branching (FB), molecular weight ($M_n$), secondary structure, and solubility, but defining the effect of these structural parameters on their biological function is challenging[11]. This is partly due to the use of β-glucans with different structures and from different sources (structural inconsistency in composition, branching structure and frequency, conformation, and molecular weight), difficulty in isolation and purification, and difficulty in structure determination[12]. For example, quality evaluation of clinically used lentinan samples shows poor batch-to-batch consistency and presence of impurities, such as proteins, lipids, and other sugar components[13,14]. A recent study also

[1]Department of Chemistry, Boston University, Boston, MA 02215, USA. [2]David H. Koch Institute for Integrative Cancer Research, Massachusetts Institute of Technology, Cambridge, MA 02139, USA. [3]Shenzhen Middle School, Shenzhen, GD 518001, China. [4]Department of Biomedical Engineering, Boston University, Boston, MA 02215, USA. [5]Lee Kong Chian School of Medicine, Nanyang Technological University, Singapore 308232, Singapore. [6]Tosoh Bioscience LLC, King of Prussia, PA 19406, USA. [7]These authors contributed equally: Ruiqing Xiao, Jialiu Zeng. ✉e-mail: mgrin@bu.edu

**Fig. 1** | Chemical structure of lentinan, a (1→3)(1→6)-β-glucan with two (1→6)-β-glucose branches every five glucose units in the (1→3)-β-linked main chain.

reveals that laminarin, a (1→3)(1→6)-β-glucan, purchased from different vendors can be either a Dectin-1 agonist or antagonist depending on the purity and structure[15]. The low solubility of β-glucans is another obstacle for their clinical use, as systemic administration of insoluble or particulate β-glucans can cause significant adverse health effects, such as microembolization, granuloma formation, inflammation and pain, as well as higher sensitivity to endotoxins[16].

These challenges motivate the development of synthetic methodologies to prepare water-soluble β-glucans and their analogs with chemically defined structures and batch-to-batch consistency[17–19]. Advances in carbohydrate chemistry and polymer science are enabling access to synthetic (1→6)-α-linked polysaccharides, however, the preparation of polysaccharides with other linkages, especially those with (1→3)-β- and (1→6)-β- configurations, still remains elusive[20]. For example, cationic ring-opening polymerization of 1,3-anhydro-2,4,6-tri-O-(p-bromobenzyl)-O-D-glucopyranose followed by deprotection affords an oligomeric (l→3)-β-glucan with $M_n$ of only 1200 g/mol[21]. Poly-condensation of difunctional 2,3,4-tri-O-acetyl-α-D-glucopyranosyl bromide also results in (1→6)-β-glucan oligomers, although a recent study shows that microwave irradiation improves the glycosylation efficiency[22]. De novo synthesis of β-glucan provides well-defined β-glucan oligomers for studying β-glucan receptor recognition[23]. However, these oligomers are too short (degree of polymerization, DP, ≤ 17) to match natural β-glucan in binding affinity or biological activities[24]. The de novo syntheses, either iteratively or convergently, are usually tedious and involve more than 10 or even 20 step reactions, which significantly limits the synthetic scale and overall yield[25–27]. Furthermore, the de novo synthesis of β-glucan with (1→6)-β-branches still remains difficult and not thoroughly investigated given the limited number of reports over the last decades[28].

Due to these challenges, replacing the natural O-glycosidic linkage in polysaccharides with other linkages, such as amide, carbonate, ester, and orthoester linkages, represents a promising strategy to synthesize polysaccharide mimetics with defined structures and high reproducibility[29–35]. Our interest focuses on the synthesis of biologically active polysaccharide mimetics inter-connected by amide linkages, which are termed poly-amido-saccharides (PASs)[36,37]. Although PASs are (1→2)-α-amide linked, the length of each repeating unit in the polymer backbone (-C1-C2-CO-NH-) is similar to that of the (1→3)-O-linked β-glucans (-C1-C2-C3-O-). The rigid glucopyranose backbone of PASs mirrors the rigid backbone of (1→3)-β-glucans. Herein, we describe the first synthesis of novel (1→6)-β-glucose branched PASs by anionic ring-opening polymerization of a gentiobiose-derived disaccharide β-lactam (Gen-lactam) followed by post-polymerization debenzylation. Via copolymerization of Gen-lactam with a glucose-based β-lactam (Glc-lactam), we also prepare a series of (1→6)-β-glucose branched PASs with different FB and $M_n$ values. Possessing pendant (1→6)-β-D-glucose branches along the amide-linked rigid glucopyranose backbone, these polymers mimic the branching structure of natural (1→3)(1→6)-β-glucans. Exposure of RAW264.7 macrophages to (1→6)-β-glucose branched PASs leads to

enhanced secretion of tumor necrosis factor (TNF-α) and nitric oxide (NO) in a FB-, dose-, and $M_n$-dependent manner. The branched PAS with a FB value of 30% elicits a potent NF-κB/AP-1 response in RAW-Blue cells via activation of the Dectin-1 receptor. This lead branched PAS also induces M1 macrophage polarization in THP-1 cells as well as primary human macrophages derived from peripheral blood mononuclear cells (PBMCs).

## Results and discussion

### Monomer design and synthesis

Due to the difficulty in constructing the (1→3)-β-linked D-glucopyranose backbone, the conventional strategy to prepare β-glucan mimetics focuses on glycosylation of linear polysaccharides[20]. Introduction of (1→6)-β-sugar branches to linear polysaccharides, such as curdlan, cellulose, and chitin, significantly increases their antitumor and immuno-modulatory activities[38–40]. Although being versatile, this glycosylation strategy suffers from irregular and inefficient glycosylation, limited solubility of polysaccharides in organic solvent, and severe degradation of the backbone[41–43]. For example, introduction of N-acetyl-glucosamine branches to a curdlan with M of 356,000 g/mol affords a branched polysaccharide of only 9000 g/mol after repeated protection/deprotection and glycosylation reactions[17]. The polymerization of disaccharide monomers with a pre-installed (1→6)-β-sugar branch would offer better control over the polymer regio- and stereo-regularity, branching structure, and frequency of branching, as well as molecular weight. Therefore, in order to circumvent the limitations of conventional methods to prepare β-glucan mimetics, we undertook an approach that combined the preparation and polymerization/copolymerization of a disaccharide β-lactam monomer with an efficient post-polymerization deprotection. Central to our approach is the anionic ring-opening polymerization of the (1→6)-β-linked disaccharide β-lactam, a technique we previously employed to synthesize linear PAS polymers with high molecular weights, low dispersities, and high stereoregularity[36].

We synthesized the disaccharide monomer, Gen-lactam, via the [2 + 2] cycloaddition of 3,4,2′,3′,4′,6′-hexa-O-benzyl-D-gentiobial with trichloroacetyl isocyanate followed by in situ removal of the N-tri-chloroacetyl group (Supplementary Fig. 1)[44]. Diastereopure Gen-lactam was obtained as a white solid after flash silica chromatography and recrystallization. We confirmed the structure of Gen-lactam by [1]H NMR, [13]C NMR, COSY, HSQC, and HMBC combined with FTIR spectroscopy, and electrospray ionization mass spectrometry (ESI-MS) analyses (Supplementary information).

### Homopolymer synthesis and characterization

With the Gen-lactam monomer prepared, we first studied its homopolymerization in tetrahydrofuran (THF), using lithium bis(-trimethylsilyl)amide (LiHMDS) as catalyst and 4-nitrobenzoyl chloride (4-NO₂BzCl) as initiator (Fig. 2)[45]. We varied the monomer-to-initiator ([M]/[I]) ratios from 15:1 to 100:1 in order to synthesize benzylated homopolymers (P1′-P4′) with different DP values. All the polymerization reactions proceed smoothly with high monomer conversions (>85%) (Supplementary Fig. 2). Gel permeation chromatography (GPC) analysis, with THF as eluent and polystyrene as standards, shows that the benzylated polymers (P1′-P4′) have $M_n$ up to 64,300 g/mol with narrow dispersities (Đ = 1.04–1.07) (Table 1 and Fig. 3a), which is considerably higher than the $M_n$ values of polymers obtained from cationic ring-opening polymerization of benzylated 1,6-anhydrodisaccharides (maximum $M_n$: 23,000 g/mol)[46].

The benzylated polymers were successfully deprotected using sodium metal (Na) in liquid ammonia [NH₃ (l)], and purified by exhaustive dialysis against water. After lyophilization, deprotected Gen-PASs (P1-P4) were obtained as fluffy white solids in high yields (81–93%). Removal of the benzyl protecting groups from the polymers was quantitative, as confirmed by disappearance of the aromatic

**Fig. 2 | Gen-PASs homopolymer synthesis and deprotection.** Reagents and conditions: **a** 4-NO₂BzCl, LiHMDS, THF, 0 °C, yield: 82–90%; **b** Na, NH₃ (*l*), −60 °C, yield: 81–93%.

proton peaks at 6.80–7.43 ppm in the $^1$H NMR spectra and of the aromatic C-H stretch at 3000–3100 cm$^{-1}$ in the FTIR spectra (Supplementary Figs. 3 and 4). The $^1$H NMR spectra of P1-P4 collected in D₂O show well-resolved signals and couplings (Supplementary Fig. 3). In the $^{13}$C NMR spectra, the carbon signals corresponding to the disaccharide repeating unit appear as thirteen sharp peaks, indicating that Gen-PASs are stereoregular polymers and no epimerization occurred under the polymerization and deprotection conditions (Supplementary Fig. 5). Based on aqueous GPC analysis with dextran standards, polymers P1-P4 possess $M_n$ values ranging from 7500 to 36,700 g/mol with narrow dispersities (Đ = 1.16–1.24). The DP$_{(GPC)}$ values of P1-P4 are higher than the corresponding DP$_{(GPC)}$ values of P1'-P4' (Table 1). The discrepancy in DP$_{(GPC)}$ values before and after deprotection may arise from underestimation of the molecular weight of benzylated polymers (Supplementary Table 1). Thus, we synthesized Gen-PASs using *tert*-butylacetyl chloride as the initiator. $^1$H NMR spectroscopy end-group analysis confirmed that the DP$_{(NMR)}$ values of benzylated polymers are consistent with DP$_{(NMR)}$ values of the deprotected Gen-PASs (Supplementary Fig. 6 and Supplementary Table 2).

Due to the adverse side effect of insoluble and particulate β-glucan, water soluble β-glucan and mimetics are more favorable for biomedical and therapeutic applications[16]. Fortunately, all the Gen-PASs are soluble in water, but the solubility decreases with increasing chain length: polymers P1 and P2 are highly water-soluble at concentrations >60 mg/mL, while polymers P3 and P4 exhibit a solubility limit of approximately 46 and 17 mg/mL, respectively. This phenomenon is consistent with our previous observation that high molecular weight glucose-derived PASs, Glc-PASs, show reduced water solubility compared to those with low $M_n$ values[36]. However, unlike Glc-PASs which tend to precipitate from solutions over time, the solutions of Gen-PASs are stable and remain clear after a week of observation at 25 °C, possibly because the pendant (1→6)-β-glucose branches in Gen-PASs prevent intermolecular aggregations of polymer chains[47].

Next, we investigated the secondary structure of Gen-PASs with circular dichroism (CD) in aqueous solutions. As evident in Fig. 3c, the CD spectra of P1-P4 display a minimum at 222 nm and a maximum at 188 nm, similar to the CD spectra of Glc-PASs, indicating that Gen-PASs adopt a similar helical conformation in aqueous solutions[48]. To evaluate the stability of this conformation, we recorded the CD spectrum of P3 under various conditions. As shown in Fig. 3d, the CD spectrum remains almost unchanged over a broad pH range from 2.0 to 12.0, or in presence of high ionic strength (2.0 M NaCl), or with protein denaturant (4.0 M urea); increasing the temperature from 5 to 75 °C leads to a modest decrease in CD intensity. In general, the conformation of Gen-PASs is stable in aqueous solutions.

### (1→6)-β-Glucose branched PASs with different FB values

The FB value of polysaccharides plays important roles in determining their physicochemical and biological functions, such as lectin recognition, antitumor, anti-HIV, or anti-coagulant activities[49,50]. An attractive advantage of the polymerization strategy is that the FB value is readily tuned by changing monomer feed ratio of disaccharide and monosaccharide monomers in copolymerization[50]. To prepare branched PASs with different FB values, we explored the copolymerization of Gen-lactam and Glc-lactam by varying the feed mole fractions of Gen-lactam from 0 to 70% (Fig. 4) and obtained a series of PAS polymers with similar molecular weights (Table 2, Fig. 3e, and Supplementary Fig. 7). After removal of the benzyl protecting groups, the proton signals of the deprotected copolymers P5-P9 are well resolved to enable compositional analysis (Supplementary Fig. 8). As shown in Table 2, the calculated FB values are in good agreement with the feed mole fractions of Gen-lactam. All the deprotected copolymers P5-P9 are soluble in water, and solubility increases with increasing FB values[47]. As before, CD analysis shows that copolymers P5-P9 adopt a helical conformation and the spectra are similar to the homopolymers P1-P4 in aqueous solutions (Fig. 3f).

### In vitro cytotoxicity of (1→6)-β-glucose branched PASs

Desirable immunomodulators should effectively potentiate the host immune system while not being cytotoxic. To assess whether (1→6)-β-glucose branched PASs are cytotoxic, polymer solutions with different concentrations were added to the culture medium of RAW264.7 macrophages, liver hepatocellular carcinoma cells (HepG2), and murine sarcoma cancer cells (Sarcoma-180), and incubated for 24 h. We measured the relative viabilities of polymer-treated cells using a MTS assay with normalization to the medium-treated controls. As shown in Fig. 5a, none of the branched PASs inhibit proliferation of RAW264.7 cells at concentrations up to 1000 μg/mL. Additionally, the polymers are also non-cytotoxic to HepG2 and Sarcoma 180 cells, as the viabilities are similar to the medium-treated controls (Fig. 5b, c).

### Table 1 | Polymer Characterization using GPC, NMR, and Optical Rotation

| Entry | $M_{n(theo)}$[a] | $M_{n(GPC)}$[b] | DP$_{(GPC)}$[b] | Đ[c] | [α]0ex25D[d] | Yield[e] |
|-------|-----------|----------|---------|------|------------|---------|
| P1' | 13,530 | 13,300 | 13.7 | 1.06 | +47.9 | 90 |
| P2' | 22,450 | 20,700 | 22.0 | 1.04 | +45.8 | 94 |
| P3' | 44,750 | 41,100 | 44.9 | 1.05 | +46.1 | 89 |
| P4' | 89,350 | 64,300 | 70.9 | 1.07 | +46.6 | 82 |
| P1 | 5420 | 7500 | 19.9 | 1.18 | +63.9 | 81 |
| P2 | 8920 | 10,900 | 29.5 | 1.20 | +68.0 | 84 |
| P3 | 17,700 | 19,200 | 53.2 | 1.16 | +67.3 | 86 |
| P4 | 35,250 | 36,700 | 103.0 | 1.24 | +65.8 | 93 |

[a]Calculated based on [M]/[I] ratios, g/mol.
[b]Determined by THF GPC against polystyrene standards for P1'-P4', or aqueous GPC against dextran standards for P1-P4, g/mol.
[c]$M_w/M_n$.
[d]Measured by polarimeter in chloroform for P1'-P4' or water for P1-P4, °.
[e]Isolated yield, %.

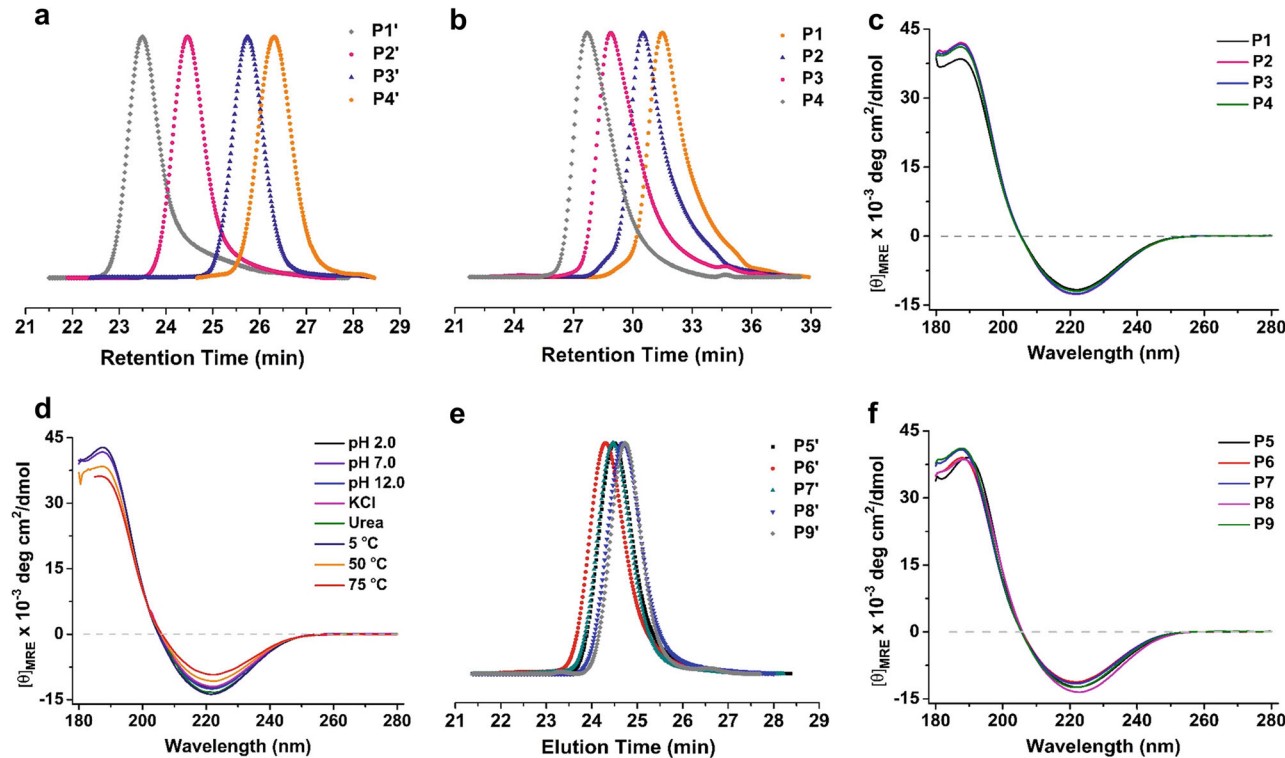

**Fig. 3 | GPC and CD characterization of PAS polymers. a** THF GPC traces of polymers P1'-P4'; **b** Water GPC traces of polymers P1-P4; **c** CD spectra of polymers P1-P4 in water at 25 °C; **d** CD spectra of polymer P3 in water under various conditions; **e** THF GPC traces of polymers P5'-P9'; **f** CD spectra of polymers P5-P9 in water at 25 °C.

**Fig. 4 | Synthesis of (1→6)-β-glucose branched PASs by copolymerization of Gen-lactam with Glc-lactam.** Reagents and conditions: **a** 4-NO₂BzCl, LiHMDS, THF, 0 °C, yield: 84–91%; **b** Na, NH₃ (l), −60 °C, yield: 80–92%.

Prior to the immunomodulatory activity evaluation, we also determined the endotoxin level of the PASs using the Limulus Amebocyte Lysate (LAL) assay in endotoxin-specific (ES) buffer to eliminate cross-reactivity with β-glucan reagents (Supplementary Fig. 9)[51]. All the polymers show endotoxin levels <0.06 EU/mL in endotoxin specific buffer and are considered endotoxin free. Notably the natural glucans all exhibit higher signal in standard LAL reagent water compared to ES buffer. Under the conditions of our assay the curdlan signal exceed our threshold of 0.06 EU/mL due to the cross-reactivity of β-glucans in this assay.

**Immunomodulatory activities of (1→6)-β-glucose branched PASs**
It is well documented that β-glucans activate the host immune system and raise the functional activities of various innate immune cells[52,53]. Macrophages represent an important component of the first line in protecting the body against foreign substance and invading pathgens[54]. Macrophages express typical cell surface receptors called pattern recognition receptors (PRRs) that recognize the β-glucan

components of fungi and bacteria, such as Dectin-1, complement receptor 3 (CR3), scavenger receptors (SRs), and Toll-like receptors (TLRs)[55]. Binding of β-glucans to these PRRs on the surface of macrophages induces the activation of transcription factors such as NF-κB family, which subsequently mediates elevated expression of pro-inflammatory cytokines and mediators, such as TNF-α, interleukins, and nitric oxide (NO)[56,57]. Therefore, enhanced cytokine production is often used as a key indicator of macrophage activation[58,59].

**TNF-α secretion**
TNF-α is one of the most important immunomodulatory cytokines and performs a pivotal part in pleiotropic effects, such as tumor regression, inflammation, angiogenesis, and immunity[60]. Upon appropriate stimulation, macrophages secrete TNF-α, which further activates macrophages in an autocrine manner, leading to a more efficient attack on foreign invaders and malignant cells[61]. To assess the effect of (1→6)-β-glucose branched PASs on TNF-α secretion, we treated RAW264.7 macrophages with various polymers at a concentration of

**Table 2 | Characterization of PAS copolymers using GPC, NMR, and Optical Rotation**

| Entry | Gen-lactam feed ratio | $M_{n(GPC)}$[a] | Đ[b] | $[\alpha]0ex25D$[c] | FB[d] | Yield[e] |
|---|---|---|---|---|---|---|
| P5' | 0 | 37,200 | 1.05 | +82.5 | N.D. | 88 |
| P6' | 10 | 41,400 | 1.05 | +78.6 | N.D. | 91 |
| P7' | 30 | 38,600 | 1.06 | +72.8 | N.D. | 90 |
| P8' | 50 | 35,800 | 1.08 | +64.1 | N.D. | 87 |
| P9' | 70 | 34,600 | 1.04 | +58.1 | N.D. | 84 |
| P5 | 0 | 17,500 | 1.34 | +152.0 | 0 | 80 |
| P6 | 10 | 21,100 | 1.37 | +137.6 | 10.3 | 89 |
| P7 | 30 | 18,600 | 1.36 | +114.7 | 31.2 | 86 |
| P8 | 50 | 17,700 | 1.47 | +87.9 | 49.5 | 92 |
| P9 | 70 | 17,400 | 1.30 | +78.3 | 71.8 | 85 |

[a]Determined by THF GPC against polystyrene standards for P5'-P9', or aqueous GPC against dextran standards for P5-P9, g/mol.
[b]$M_w/M_n$.
[c]Measured by polarimeter in chloroform for P5'-P9' or water for P5-P9, °.
[d]Determined from 1H NMR spectra; N.D., not determined.
[e]Isolated yield, %.

100 µg/mL for 8 h. As depicted in Fig. 5d, all the branched PASs significantly increase the TNF-α production compared to the medium-treated control. The amount of TNF-α secretion is substantially influenced by FB values, and the branched PAS with a FB value of 30% (P7) is superior to the other polymers. In contrast, linear Glc-PAS (P5, FB = 0%) only slightly enhances TNF-α secretion, indicating that the presence of (1→6)-β-glucose branches is important for their immunomodulatory activities. For reference, we also examined the activity of laminarin (Lam), a naturally occurring water-soluble (1→3)-β-glucan with low frequencies of (1→6)-β-glucose branches[15], and detected only a moderate TNF-α secretion compared to P7 (Fig. 5d). Upon identifying the TNF-α-inducing properties of PASs, we selected the top-performing P7 to study the influence of polymer concentration on macrophage activation. As shown in Fig. 5e, P7 augments TNF-α production of RAW264.7 cells in a dose-dependent manner. The level of TNF-α increases with increasing P7 concentrations over the range of 0.1–500 µg/mL, and reaches a plateau at concentrations higher than 500 µg/mL.

## NO production
NO is a short-lived radical synthesized by the inducible NO synthase (iNOS) during macrophage activation[62]. Accumulating evidence shows that NO is a crucial messenger mediating diverse biological functions, such as neuronal transmission, vascular relaxation, immune modulation, and cytotoxicity against microorganisms and malignant cells[63]. To evaluate the NO-inducing activity of PASs, we treated RAW264.7 cells with polymers at a concentration of 100 µg/mL for 24 h. As shown in Fig. 5f, the level of NO produced from the P5-treated cells is very low, indicating its weak stimulating activity. On the other hand, NO production increases after incubation with (1→6)-β-glucose branched PASs. Treatment with P7 yields the greatest amount of NO, which is 16.8 times that of the control group. For comparison, Lam-treated cells only secrete a moderate level of NO compared to P7. Next, we examined whether polymer concentration affects NO production. As shown in Fig. 5g, a steady increase in NO secretion occurs with increasing concentration of P7 from 0.1 to 1000 µg/mL. At the highest concentration studied, the NO level is 32.6 times that of the medium-treated cells.

## Influence of $M_n$ on TNF-α and NO secretion
Molecular weight is another important factor that influences the biological activities of polysaccharides[23,64]. In order to investigate the influence of $M_n$ on the immunomodulatory activities of

branched PASs, we synthesized a series of 30% branched PASs (P10-P13) with $M_n$ values ranging from 3600 to 46,100 g/mol (Supplementary Table 3 and Supplementary Fig. 10). With these polymers in hand, we evaluated their TNF-α- and NO-inducing activities under the same conditions as previous studies. As depicted in Fig. 5h, the TNF-α secretion increases with increasing $M_n$, and then reaches a plateau at the $M_n$ value of 18,600 g/mol (P7). Further increase of the $M_n$ does not significantly enhance TNF-α production. On the other hand, the NO production steadily increases over the entire $M_n$ range (Fig. 5i). Treatment with polymer P13 ($M_n$ = 46,100 g/mol) substantially increases the secretion of NO by 24.4 times compared to the medium-only control.

## Activation of Dectin-1/NF-κB pathway
Having identified that (1→6)-β-glucose branched PASs stimulate macrophages, we investigated whether the activation is mediated by the NF-κB pathway and whether Dectin-1 is involved in this process as with natural β-glucans. We used the RAW-Blue reporter cell line (derived from RAW264.7 cells) that stably expresses secreted alkaline phosphatase (SEAP) upon NF-κB and activator protein 1 (AP-1) transcriptional activation[65]. RAW-Blue cells express Dectin-1, the major β-glucan receptor recognizing both (1→3)-β-glucans and (1→3)(1→6)-β-glucans[66,67]. As shown in Fig. 6a, incubation of RAW-Blue cells with Dectin-1 agonists, such as curdlan [Cur, a (1→3)-β-glucan], Lam [a (1→3)(1→6)-β-glucan], or Lentinan [Lent, a (1→3)(1→6)-β-glucan] significantly enhances the NF-κB activation, as seen by the increased SEAP response[68]. The response upon treatment with Lam is consistent with a recent finding that the immunomodulatory activities of Lam are highly dependent on its purity, and purified samples afford greater biological effects[15]. On the other hand, dextran (Dex), a (1→6)-α-glucan, fails to activate the NF-κB/AP-1 pathway in macrophages, as it is not recognized by PRRs on the cell surface[66]. Linear Glc-PAS (P5), which possesses an (1→2)-α-amide-linked glucopyranose backbone but no β-(1→6) glucose branches, only elicits a minimal increase in SEAP expression, in accordance with its weak TNF-α and NO-inducing activities. These results indicate that the (1→2)-α-amide bond of Glc-PAS may not be a favorable structural attribute for PRR recognition compared to the (1→3)-β- and (1→6)-β- linkages of natural β-glucans[64]. On the other hand, treatment of RAW-Blue cells with P7 affords stronger NF-κB/AP-1 macrophage activation relative to that of the P5-exposed cells. Neutralization of Dectin-1 with anti-Dectin-1 antibody significantly decreases the SEAP response elicited by polymer P7 to basal expression levels (Fig. 6b), indicating that activation of NF-κB in RAW-Blue cells depends on the Dectin-1 receptor. Therefore, the presence of (1→6)-β-glucose branches is critical for the Dectin-1 recognition and macrophage activation by the branched PASs, and the (1→2)-α-amide-linked PAS backbone acts as a scaffold for presenting (1→6)-β-glucose branches to PRRs on macrophages. These findings are consistent with previous reports in which introduction of (1→6)-β-sugar branches to cellulose and chitin [β-(1→4)-glucans] enhances their antitumor and immunomodulatory activities[38,40], and that presence of β-(1→6)-glucose branches increases the recognition of β-glucan by Dectin-1[64]. Addition of polymyxin B (PMB), a LPS inhibitor, does not attenuate the SEAP response elicited by the PAS polymers, confirming that the macrophage activation is not due to LPS contamination (Supplementary Fig. 11).

## Macrophage polarization of (1→6)-β-glucose branched PASs
Polysaccharides play an important role in macrophage polarization into either classically activated (M1; pro-inflammatory) or alternatively activated (M2; anti-inflammatory) macrophages[69–71]. We chose the lead polymer P7, which induces the highest level TNF-α and NO, to investigate its effect on macrophage polarization in PMA activated THP-1 cells, and compared it to the cell medium-treated control and Lam-treated cells. We determined the ability of the

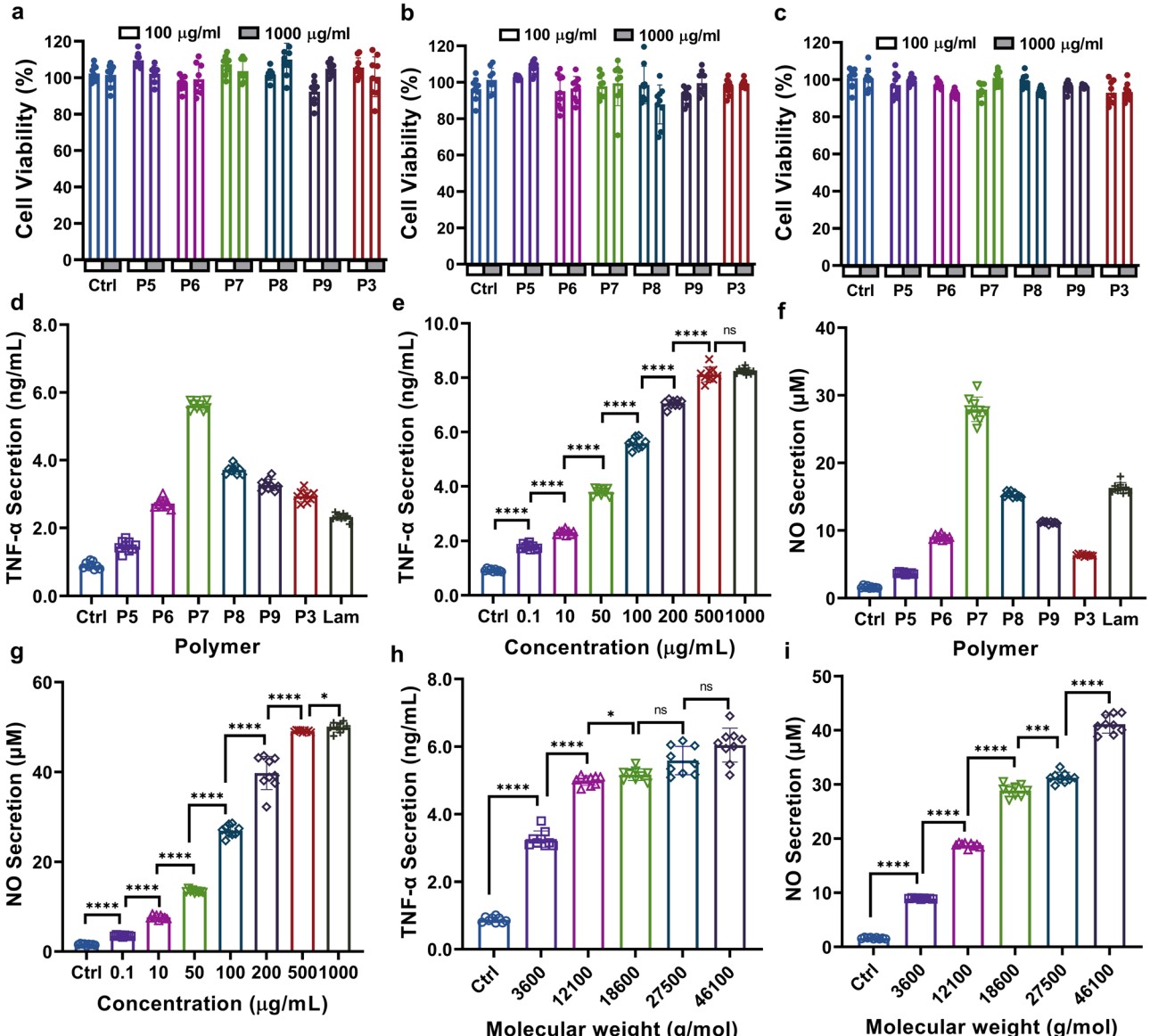

**Fig. 5 | Cytotoxicity and immunomodulatory activities of (1→6)-β-glucose branched PASs.** Cytotoxicity in **a** RAW264.7; **b** HepG2; and **c** Sarcoma−180 cell lines: 37 °C, 24 h. **d** TNF-α secretion, polymer concentration: 100 µg/mL, 37 °C, 8 h; **e** Effect of polymer concentration (P7, FB 30%) on TNF-α secretion: 37 °C, 8 h. Two-tailed unpaired t test were performed Ctrl vs 0.1 ($p < 0.0001$), 0.1 vs 10 ($p < 0.0001$), 10 vs 50 ($p < 0.0001$), 50 vs 100 ($p < 0.0001$), 100 vs 200 ($p < 0.0001$), 200 vs 500 ($p < 0.0001$), 500 vs 1000 ($p = 0.2078$); **f** NO secretion, polymer concentration: 100 µg/mL, 37 °C, 24 h; **g** Effect of polymer concentration (P7, FB 30%) on NO secretion, polymer concentration: 100 µg/mL, 37 °C, 24 h. Two-tailed unpaired t test were performed Ctrl vs 0.1 ($p < 0.0001$), 0.1 vs 10 ($p < 0.0001$), 10 vs 50 ($p < 0.0001$), 50 vs 100 ($p < 0.0001$), 100 vs 200 ($p < 0.0001$), 200 vs 500 ($p < 0.0001$), 500 vs 1000 ($p = 0.0346$); **h** Effect of $M_n$ (FB 30%) on TNF-α secretion, polymer concentration: 100 µg/mL, 37 °C, 8 h. Two-tailed unpaired t test were performed Ctrl vs 3600 ($p < 0.0001$), 3600 vs 12,100 ($p < 0.0001$), 12,100 vs 18,600 ($p < 0.0001$), 18600 vs 27,500 ($p = 0.0136$), 27,500 vs 46,100 ($p = 0.0521$); **i** Effect of $M_n$ (FB 30%) on NO secretion, polymer concentration: 100 µg/mL, 37 °C, 24 h. Two-tailed unpaired t test were performed Ctrl vs 3600 ($p < 0.0001$), 3600 vs 12,100 ($p < 0.0001$), 12,100 vs 18,600 ($p < 0.0001$), 18,600 vs 27,500 ($p = 0.0002$), 27,500 vs 46,100 ($p < 0.0001$). Unless otherwise stated, data are means ± SD of three independent experiments ($N = 3$), and triplicates ($n = 3$) per condition are used in each experiment. ns $p > 0.05$, *$p ≤ 0.05$, **$p ≤ 0.01$, ****$p ≤ 0.0001$ compared to control (**a**, **c**) or adjacent treatment group using two-tailed unpaired t test. Abbreviations used: Ctrl = Control.

polymers to polarize macrophages using a multiplex macrophage polarization assay, which differentiates macrophages into different subtypes based on their surface markers and cytokine signatures into either M1, M2 or tumor-associated macrophages. As shown in Fig. 6c, treatment of THP-1 cells with polymer P7 for 24 h at a concentration of 100 µg/mL elevates the gene expression of *TNF-α, IL-1β, IL-6, IL-12A, MMP9, CD68, CD86*, and *NOS2* compared to the medium-treated control, indicating that P7 induces polarization of THP-1 cells into M1 macrophages. In contrast, the levels of M2 and tumor-associated macrophage markers do not significantly change

after treatment with P7 (Fig. 6d). M1 macrophages secrete pro-inflammatory cytokines and chemokines, present antigens, and thus participate in the positive immune and anti-tumor responses[72,73]. These results are consistent with the enhanced production of pro-inflammatory markers such as TNF-α and NO via P7 treatment to RAW 264.7 cells as shown in Fig. 5. We also examined the activity of Lam under the same condition, and detected only a moderate increase in markers associated with M1 macrophage phenotype, indicating it polarizes macrophage to the M1 phenotype to a lesser extent compared to P7 (Fig. 6c).

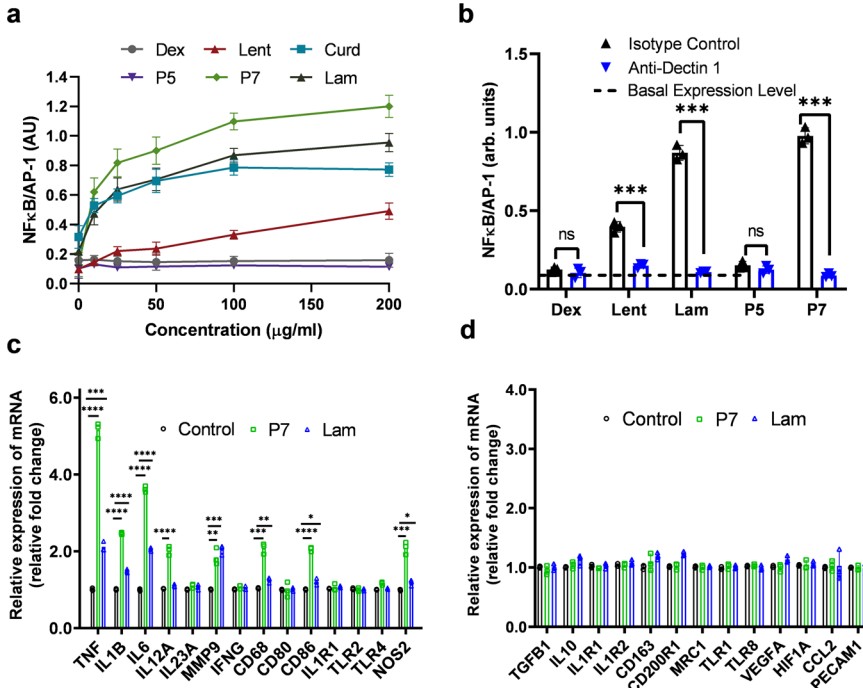

**Fig. 6 | Activation of NF-κB pathway in RAW-Blue cells and macrophage polarization in THP-1 cells. a** Dose-dependent NF-κB/AP-1 response elicited by PASs polymers, β-glucans, and dextran: 37 °C, 24 h. ($N = 2$ independent experiments, $n = 2$ per condition). **b** Effect of anti-Dectin-1 blocking antibody, or an isotype control antibody (10 μg/mL) on the NF-κB/AP-1 activation induced by P5, P7, and Lam: 100 μg/mL, 37 °C, 24 h. Two-tailed unpaired t test were performed: Lent: Isotype vs Anti-Dectin ($p < 0.0001$), Lam: Isotype vs Anti-Dectin ($p < 0.0001$), P7: Isotype vs Anti-Dectin ($p < 0.0001$). **c** M1 and **d** M2 macrophage-related marker mRNA expression in polymer-treated THP-1 cells: 100 μg/mL, 37 °C, 24 h. ($N = 3$ independent experiments, $n = 1$ per condition). Two-tailed unpaired t test were performed for M1 macrophage-related mRNA

expression: TNF: Ctrl vs P7 ($p < 0.0001$), Ctrl vs Lam ($p = 0.0001$), IL1B: Ctrl vs P7 ($p < 0.0001$), Ctrl vs Lam ($p < 0.0001$), IL6: Ctrl vs P7 ($p < 0.0001$), Ctrl vs Lam ($p < 0.0001$), IL12A: Ctrl vs P7 ($p < 0.0001$), MMP9: Ctrl vs P7 ($p = 0.0026$), Ctrl vs Lam ($p = 0.001$), CD68: Ctrl vs P7 ($p = 0.0002$), Ctrl vs Lam ($p = 0.006$), CD86: Ctrl vs P7 ($p < 0.0001$), Ctrl vs Lam ($p = 0.0165$), NOS2: Ctrl vs P7 ($p = 0.0003$), Ctrl vs Lam ($p = 0.0152$). Unless otherwise stated, data are means ± SD of three independent experiments ($N = 3$), triplicates ($n = 3$) per condition were used. ns $p > 0.05$, *$p ≤ 0.05$, **$p ≤ 0.01$, ***$p ≤ 0.001$, ****$p ≤ 0.0001$ compared to control using two-tailed unpaired t test. Abbreviations used: Ctrl = Control, Dex = Dextran, Lent = Lentinan, Lam = Laminarin.

## Polarization of primary human macrophages derived from PBMCs

While THP-1 cells have been widely used to study the function of monocytes, the extent to which THP-1 cells mimic monocytes is still not fully characterized[74]. Therefore, we further assessed the macrophage polarization activity of branched PASs in primary human macrophages[75]. We cultured CD14+ enriched M0 macrophages, isolated from peripheral blood mononuclear cells (PBMCs), with macrophage colony stimulating factor (M-CSF). After 5 days of differentiation, greater than 90% of cells expressed CD68, indicating differentiation from monocytes to macrophages (Supplementary Fig. 12). We subsequently exposed these primary M0 macrophages to media only, IL-4 (20 ng/uL), LPS (100 ng/uL), or polymers P5, P7, Lam, or Lent at a concentration of 100 μg/mL, and assessed the surface expression of CD80, CD163, and CD206[75,76]. M1-polarized macrophages exhibit significantly higher CD80 expression, moderately lower CD206 expression, and moderately higher CD163 expression compared to M2-polarized macrophages[75]. LPS, P5, P7, Lam, and Lent treatment leads to higher CD80 expression compared to the medium-treated samples and M2-polarized macrophages (IL-4) (Fig. 7a). Treatment with P7 induces significantly higher CD80 expression compared to P5, Lam, or Lent. The CD80 expression between P7 and LPS samples are not statistically different. We observed a similar trend in percentage CD80+. P7 and LPS are not statistically different, but there is a statistically significant decrease in CD80+ population with treatment of P5 ($p < 0.0001$) (Fig. 7b). Consistent with the literature, M1-polarized macrophages exhibit lower CD206 expression compared to M2-polarized macrophages[75] Compared to the unstained control,

M2-polarized macrophages display a 21.3-fold increase in CD206 expression compared to 11.6-fold increase among M1-polarized macrophages via treatment with P7, P5, Lent, and Lam (Fig. 7c, d). Finally, CD163 expression profile after treatment with P7, P5, Lent, and Lam are consistent with M1-polarized macrophages (Fig. 7e, f). All groups are significantly different from IL-4 treated, M2-polarized macrophages and untreated M0 macrophages.

Polymer P7 exhibits polarization that is not statistically different from LPS in CD80, CD206, and CD163 expression while polymer P5 shows statistically significant differences in CD80 fold change and percent CD80+. P5-treated macrophages are 66.2% CD80+ while P7 and LPS are 95.0% and 92.4% CD80+, respectively. Treatment with P5 also affords a higher percentage of CD163+ cells compared to P7 and LPS (57.0% vs. 47.1% and 44.9%, respectively). Finally, treatment with P5 exhibits similar CD206 expression relative to P7 and LPS (79.9% vs. 82.0% and 82.2%, respectively). Therefore, while P5 polarizes primary human macrophages, it does not induce as strong a response as polymer P7 and is statistically different from LPS. These results are consistent with our previous findings with macrophages, confirming the critical role of β-(1→6) glucose branches in the immunomodulatory activities of PAS polymers. Further, treatment with P7 also compares favorably in M1 polarization activity to known natural β-glucans Lent and Lam at equal concentration. P7 exhibits significantly higher CD80 activity than Lent and Lam (Fig. 7a, b). P7, P5, Lent, and Lam all exhibit statistically similar CD206 and CD163 levels compared to LPS (Fig. 7c–f). Supplementary Fig. 13 shows representative flow cytometric plots of CD80, CD206, and CD163 expression in each treatment group which support this finding.

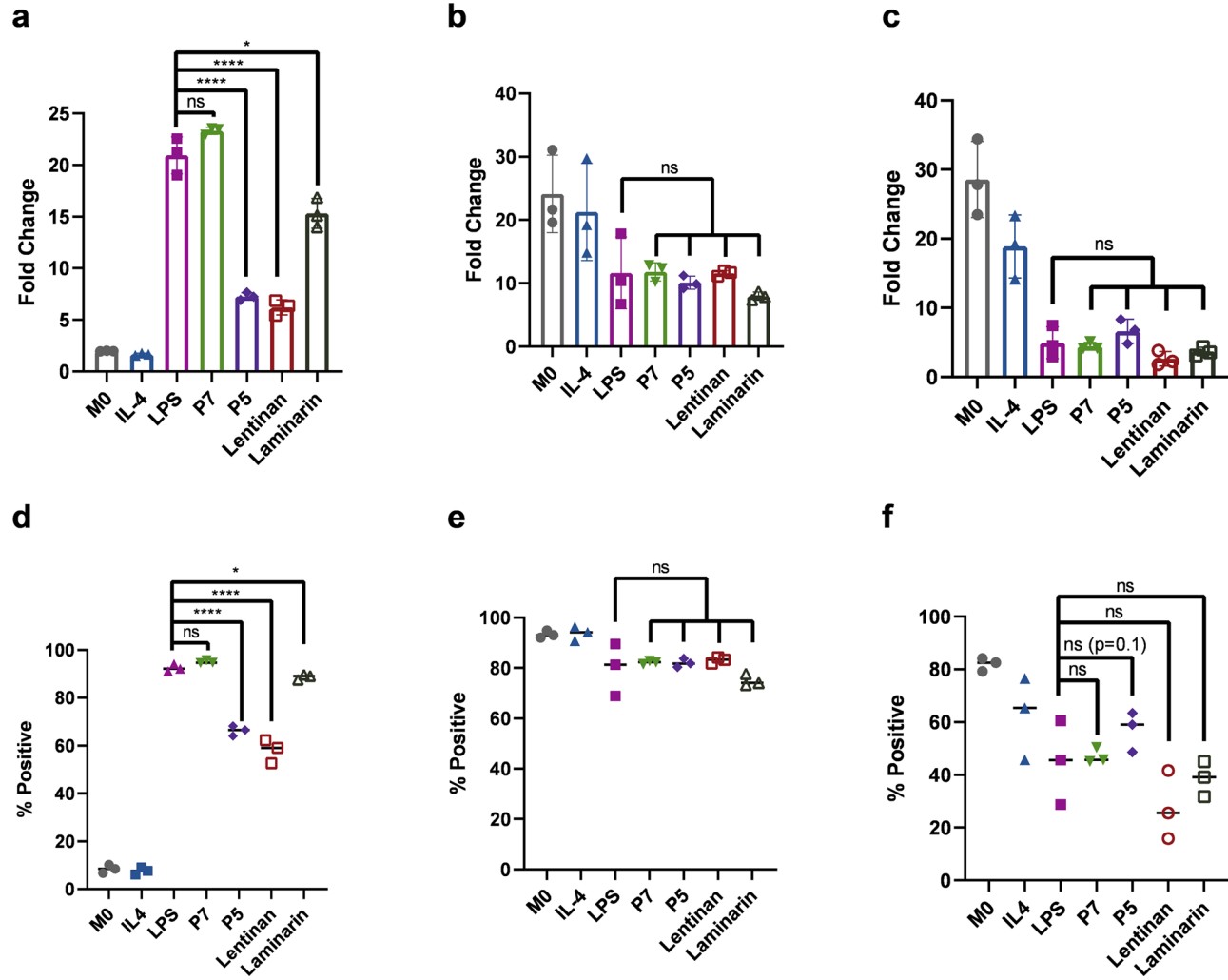

**Fig. 7 | Primary macrophage expression of M1 and M2 polarization markers evaluated by flow cytometry after treatment with medium only, IL-4, LPS, P7, P5, Lent, or Lam.** Fold change of median fluorescence intensity of **a** CD80, **b** CD206, and **c** CD163 compared to an unstained control. Two-tailed unpaired t test were performed for **a** LPS vs P5 ($p = 0.0002$), LPS vs Lentinan ($p = 0.0002$), LPS vs Laminarin ($p = 0.0132$). Percentage of positive cells of **d** CD80, **e** CD206, and **f** CD163 compared to an unstained control. Error bars represent standard deviation. Two-tailed unpaired t test were performed for **d** LPS vs P5 ($p < 0.0001$), LPS vs Lentinan ($p = 0.0003$), LPS vs Laminarin ($p = 0.0296$). Unless otherwise stated, data are means ± SD of three separate experiments ($N = 3$). ns $p > 0.05$, *$p < 0.05$, ****$p \leq 0.0001$ compared using two-tailed unpaired t test. Abbreviations used: Ctrl = Control, Dex = Dextran, Lent = Lentinan, Lam = Laminarin.

The new method, described herein, possesses several advantages over the glycosylation strategy in preparing synthetic (1→6)-β-branched polysaccharides including: (1) homopolymerization of Gen-lactam yields high molecular weight comb-shaped polymers with a (1→6)-β-glucose branch at each repeating unit; (2) preparation of polymers with different FB values via changing the feed ratio of disaccharide and monosaccharide monomers; (3) control of polymer length by varying the monomer-to-initiator ratio; and, (4) avoidance of epimerization or backbone degradation during the polymerization and deprotection reactions. Compared to the conventional polymerization methods to prepare β-glucans, our approach achieves higher molecular weight and better control over the stereo-regularity, as well as provides polymers with tunable sugar branches. Compared to the de novo synthesis strategy to prepare oligomeric β-glucans (≤17 mers), which requires multi-steps regioselective protection and deprotection, as well as glycosylation reactions, our polymerization method enables access to high molecular weight (1→6)-β-branched β-glucan mimetics (up to 46,100 g/mol) with higher efficiency (2-step reactions from β-lactam monomers) and at a larger scale (up to 500 mg of PASs from a single polymerization-and-deprotection protocol), along with the ability to control the degree of polymerization.

The MTS proliferation assays reveal that these novel β-glucan mimetics are non-cytotoxic to RAW264.7, HepG2, and Sarcoma-180 cells at a dose of 1000 µg/mL. Branched PASs stimulate mouse macrophages and enhance the production of TNF-α and NO in a FB-, dose-, and $M_n$-dependent manner. This stimulation proceeds via activation of the NF-κB/AP-1 pathway in a Dectin-1-dependent manner similar to natural β-glucans. The transcription factor NF-κB regulates multiple aspects of innate and adaptive immune functions, and plays a pivotal role in modulating inflammatory responses[77]. The PAS with a FB value of 30% is the most efficacious, consistent with previous studies where β-glucans with FB values between 20–33% exhibit the highest anti-tumor activity against Sarcoma 180 in vivo[5]. Consistent with literature, high molecular weight PASs are more immunomodulatory than polymers with lower $M_n$ values[78]. Additionally, the 30% branched PAS induces M1 macrophage polarization in THP-1 cells through increasing the gene expression of *TNF-α, IL-1β, IL-6, IL-12A, MMP9, CD68, CD86, and NOS2*, and in primary human macrophages (higher CD80, lower CD206 expression), a phenotype which exhibits pro-inflammatory and anti-tumor responses in different cancer models[72,73].

The (1→6)-β-glucose branched PASs are water-soluble polysaccharide mimetics and share structural features with natural β-

glucans, such as having a stereochemically defined glucopyranose backbone and possessing (1→6)-β-glucose branches. The aqueous solubility of PASs is advantageous as adverse side effect arise because of the insoluble and particulate nature of β-glucan. Another advantage of our synthetic method is that PASs possessing reactive terminal groups, synthesized by employing functional initiators, will facilitate conjugation of these polymers to proteins, other polymers, or particles, which, potentially, enables targeted delivery of PASs, increasing their biological activity, and decreasing doses for efficacy[79–82]. A number of successful strategies are being pursued or contemplated to overcome the solubility/particulate issues. For example, Bertozzi et al. conjugate glycopolypeptides to 0.8 μm polystyrene beads, and this colloidal suspension elicited a greater pro-inflammatory response than natural β-glucan curdlan[18]. Our laboratory and others develop local drug delivery using an implantable device that provides local high doses to increase concentration at the site of action and eliminates the need for intravenous administration[83–87]. Finally, at the nanoscale, several groups describe β glucan modified nanoparticles for targeting and delivery of active agents[80–82,88].

In this work, ring-opening polymerization of Gen-lactam and its copolymerization with Glc-lactam afford a new type of biologically active (1→6)-β-glucose PASs as β-glucan mimetics. From a clinical perspective, these (1→6)-β-glucose branched PASs are unique immunomodulators worthy of further study. Our future interest is in the treatment of sarcomas with these (1→6)-β-glucose branched PASs, given the immunomodulation activity and the efficacy precedent of β-glucans in an in vivo sarcoma model. From a polymer chemistry perspective, synthetic methodologies that afford chemically defined, stereoregular branched saccharide polymers are of significant utility as it enables the elucidation of structure-property relationships to guide the development of synthetically derived biologically-active materials.

## Methods
### Materials and instrumentation
Materials and chemicals were purchased from Sigma Aldrich or Alfa Aesar, and were used as received unless otherwise noted. 1,2,3,6,2′,3′,4′,6′-Octa-O-acetyl-α-D-gentiobiose and 3,4,6-tri-O-benzyl-D-glucal starting materials were purchased from Carbosynth, LLC (San Diego, California). Laminarin was purchased from Sigma–Aldrich and purified by extensive dialysis and lyophilization. Solvents used for the polymerization reactions were dried and freshly distilled prior to use. All reactions were carried out under nitrogen using standard techniques, unless otherwise noted. Lyophilization was performed using a Virtis Benchtop 4 K freeze dryer Model 4BT4K2L-105 at −40 °C. $^1$H NMR and $^{13}$C NMR spectra were recorded on a Varian INOVA 500 MHz spectrometer. Infrared spectroscopy (FTIR) was performed on a Nicolet FTIR with a horizontal attenuated total reflectance (ATR) adapter plate. Specific optical rotations were determined at 25 °C using a Rudolph Autopol II polarimeter operating at 589 nm in a 50 mm pathlength cell. The molecular weights of benzylated polymers were determined by gel permeation chromatography (GPC) against polystyrene standards using THF as the eluent at a flow rate of 1.0 mL/min through two Jordi columns (Jordi Gel DVB 10$^5$ Å and Jordi Gel DVB 10$^4$ Å, 7.8 × 300 mm) at 25 °C with a refractive index detector. The molecular weights of P2′ and P3′ were also measured with triple-detection GPC combining refractive index detector (RI), light scattering detector (LS), and viscometer (VISC) (Tosoh EcoSEC Elite Model HLC-8420), and the data is shown in Supplementary Table 2. The molecular weights of the deprotected polymers were determined by GPC versus dextran standards using aqueous buffer (0.2 M NaNO$_3$, 0.010 M phosphate buffer, pH 7.5) as the eluent at a flow rate of 0.50 mL/min through two Agilent PL aquagel columns (OH MIXED-M and OH 30, 7.5 × 300 mm) at 25 °C with a refractive index detector. Circular dichroism (CD) studies were performed in a 1 mm path length cuvette using an Applied Photophysics CS/2Chirascan with a standard Mercury lamp.

### General procedure for polymerization
The procedure was adapted from the literature with minor changes[36]. In an oven-dried flask, Gen-lactam (0.400 g, 0.448 mmol) and p-nitrobenzoyl chloride (6.67 mol% for DP$_{(theo)}$ =15, 4.0 mol% for DP$_{(theo)}$ = 25, 2.0 mol% for DP$_{(theo)}$ = 50, and 1.0 mol% for DP$_{(theo)}$ = 100) were dissolved in 6.0 mL of freshly distilled tetrahydrofuran (THF, without BHT) which had been dried over molecular sieves. Because small quantities of initiator were required, a stock solution of p-nitrobenzoyl chloride in tetrahydrofuran was prepared immediately before use and the appropriate volume was added to the reaction flask. The reaction flask was cooled to 0 °C in an ice bath. Next, an appropriate volume of a 0.25 M solution of LiHMDS in THF (16.7 mol% for DP$_{(theo)}$ = 15, 10.0 mol% for DP$_{(theo)}$ = 25, 5.0 mol% for DP$_{(theo)}$ = 50, and 2.5 mol% for DP$_{(theo)}$ = 100) was added and the solution was stirred 2–6 h, and the reaction was monitored by GPC. To quench the reaction, a drop of sat. NH$_4$Cl aqueous solution was added to the reaction. The solvent was removed and the resultant solid was re-dissolved in dichloromethane (25 mL) and washed with 1 M HCl, sat.NaHCO$_3$, and brine. After drying over Na$_2$SO$_4$, the solvent was removed under vacuum. The product was dissolved in minimal dichloromethane and precipitated by adding dropwise into a flask of stirred, cold hexane (50 mL). The resultant solid was collected by filtration, re-dissolved in dichloromethane, and precipitated in cold methanol. The precipitate was collected by filtration and dried under high vacuum (82–94%). Spectral data is listed for P2′: $^1$H NMR (500 MHz, CDCl$_3$) δ 7.45–6.75 (br m, 30H), 5.41 (br s, 1H), 4.92–2.60 (br m, 26H); $^{13}$C NMR (125 MHz, CDCl$_3$) δ 169.4, 138.8, 138.6, 138.5, 138.1, 137.7, 128.1 (3), 127.8, 127.7, 127.4, 127.0, 95.7, 82.1, 79.0, 75.4, 74.8, 74.1, 73.2, 72.9, 72.2, 71.6, 70.8, 69.3, 68.0, 51.6. IR (ATR): 3347 br (NH), 1700 (amide I), 1496 br (amide II), 1069 cm$^{-1}$.

Copolymerization of Gen-lactam and Glc-lactam was conducted according to similar procedure.

### General procedure for deprotection
The procedure was adapted from the literature with minor changes[36]. To a polymer (0.32–0.35 g depending on the sample) solution in 5.0 mL of THF, 1.5 equivalent of LiHMDS was added and the solution was stirred for 5 min at room temperature. The solution was then added into a rapidly stirred solution of sodium metal in anhydrous liquid ammonia (100 mL) at −60 °C (dry ice-isopropyl ether bath) under nitrogen (sodium was washed in cyclohexane and cut into small pieces before addition). The solution's deep blue color was maintained by adding additional sodium. After 1 h at −60 °C, sat. NH$_4$Cl solution was added until the blue color disappeared. After evaporation of the ammonia in a warm water bath, the resulting residue was washed with diethyl ether. The aqueous solution was dialyzed at room temperature for 2 days with six water changes (Spectrum labs dialysis tubings, MWCO 2000–10,000 depending on the samples). The solutions were then lyophilized to give deprotected Gen-PASs as white fluffy solids (81–93%). Spectral data is listed for P2: $^1$H NMR (500 MHz, D$_2$O) δ 5.75 (d, J = 4.8 Hz, 1H), 4.46 (d, J = 8.0 Hz, 1H), 4.18 (t, J = 10.1 Hz, 1H), 3.95 (m, 3H), 3.72 (m, 2H), 3.48 (m, 3H), 3.39 (t, J = 9.3 Hz, 1H), 3.32 (t, J = 8.6 Hz, 1H), 3.05 (dd, J = 11.2, 4.8 Hz, 1H); $^{13}$C NMR (125 MHz, D$_2$O) δ 172.8, 105.4, 78.7, 78.4, 77.7, 75.9, 74.6, 72.4, 72.3, 71.0, 70.5, 63.6, 54.0. IR (ATR): 3350 br, 1680 (amide I), 1540 (amide II), 1032 cm$^{-1}$.

### Limulus amebocyte lysate assays
The endotoxin levels of branched PASs (P1-P13), natural glucans curdlan, laminarin, and dextran were determined by a quantitative end point assay based on the reactivity of gram-negative endotoxin with a modified Limulus Amebocyte Lysate (LAL) and a synthetic color producing substrate to detect endotoxin chromogenically at 37 °C using ToxinSensor™ Chromogenic LAL Endotoxin Assay Kit (Genscript, Cat. No. L00350). The standard endotoxin (10 EU/mL) was from E.coli provided in the kit. All the polymers showed endotoxin levels <0.06 EU/mL.

The endotoxin levels of all PAS polymers and β-glucans used in primary macrophage polarization assays were also examined at concentrations of 100 ng/mL with and without endotoxin specific assay buffer (Wako/Fujifilm, ESB-0006), which contains high concentrations of hydroxymethylated curdlan. Sensitivity of the assay was determined with and without ES buffer (Supplementary Fig. 9A), and all polymers exhibited expected results (Supplementary Fig. 9B).

## Cell culture and in vitro cell viability assay

The cytotoxicity of Gen-PASs and copolymers was evaluated using an MTS cell proliferation assay in three cell lines: murine macrophage (RAW264.7), human liver hepatocellular carcinoma (HepG2), and murine Sarcoma cancer (Sarcoma 180) cells in vitro. RAW 264.7, HepG2 and Sarcoma 180 cells (ATCC) were maintained at 37 °C under 5% $CO_2$ using DMEM media containing 10% fetal bovine serum (FBS, R & D Systems, S11150) and 1% penicillin–streptomycin. THP-1 cells (ATCC) were maintained at 37 °C under 5% $CO_2$ using RPMI media containing 10% fetal bovine serum and 1% penicillin–streptomycin. MTS assay (CellTiter 96 Aqueous One, Promega, Madison, WI) was used to assess polymer cytotoxicity. Briefly, cells were cultured in a 96-well plate at $1.0 \times 10^4$/well for 24 h, after which the medium was exchanged for media containing either medium treatment or 100 or 1000 μg/mL of polymers. The cells were incubated with treatment for 24 h, after which cell viability was quantified relative to the medium-treated control, after correcting for background absorbance at 492 nm (BioTek Synergy HT). Three wells per treatment concentration were used, and the assay was repeated three times for each cell line.

## Assay for TNF-α secretion in RAW 264.7 cells

The assay was adapted from the literature with minor changes[40]. RAW264.7 cells were seeded in each well of 24-well plates ($5 \times 10^5$/well) with 1 mL of DMEM (Gibco, 31053036) containing 10% heat-inactivated FBS. After incubating the cells for 24 h at 37 °C, the polymer solutions (final concentration 100 μg/mL) were added in triplicates, and the cells were incubated at 37 °C for 8 h. TNF-α secretion was measured using the L929 cell bioassay[89] as follows: L929 cells were seeded in each well of 96-well plates ($1.0 \times 10^4$/well) with 50 μL MEM (Gibco, 21475025) containing 10% heat-inactivated FBS. After 24 h at 37 °C, 25 μL of actinomycin D (4 μg/mL) was added to each well. A sample solution, which was obtained by dilution with MEM up to 625 times of the culture supernatant of RAW264.7 cells with the polymers, or diluted TNF-α solution (Enzo life sciences, ALX-522-009-C050 at a concentration of 0.0032-10 ng/mL in MEM) was then added (25 μL/well) to the cells. After 24 h, the medium was removed, and the cells were washed with PBS. They were fixed with 100 μL/well of glutaraldehyde solution (Sigma–Aldrich, G5882), which was obtained by diluting aqueous glutaraldehyde with PBS 100 times and washed with water. A 0.2% crystal violet (Sigma Aldrich, C0775) solution (w/v) in 2% ethanol solution (50 μL/ well) was added, and after 30 min, the cells were washed with water. The stained cells were lysed with 100 μL/well of 50% (v/v) ethanol containing 50 mM sodium dihydrogen phosphate (Sigma–Aldrich, 1.06342), and the absorbance at 595 nm was measured (BioTek Synergy HT). Three wells per treatment concentration were used, and the assay was repeated three independent times.

## Assay for NO secretion in RAW264.7 cells

The assay was adapted from the literature with minor changes[40]. RAW264.7 cells were seeded in each well of 24-well plates ($5 \times 10^5$/well) with 1 mL of DMEM containing 10% heat-inactivated FBS. After a 24 h incubation at 37 °C, polymer solutions at (final concentration 100 μg/mL) were added in triplicates, and the cells were incubated for 24 h at 37 °C. NO secretion was assayed using Griess reagent system (Promega, G2930) following manufacturer's protocol: briefly, 100 μL of the supernatant was treated with an equal volume of the Griess reagent (2% sulfanilamide, 0.2% naphthylethylene diamine dihydrochloride, and 5%

phosphoric acid) for 10 min at room temperature in the dark. The optical density of the mixture was measured at 540 nm, with sodium nitrite solutions (0.5–100 μM) being used as standards (BioTek Synergy HT). Three wells per treatment concentration were used, and the assay was repeated three independent times.

## RAW-Blue and NF-κB-dependent secreted alkaline phosphatase (SEAP) assay

NF-κB induction was quantified by measuring the levels of secreted alkaline phosphatase (SEAP) in the culture supernatant using QUANTI-Blue assay and RAW-Blue cells (InvivoGen, raw-sp) according to the manufacturer's instructions. In brief, $1.0 \times 105$ cells per well of a 96-well plate were treated with Glc-PAS, Gen-PAS-30%, Dextran, Curdlan (InvivoGen, tlrl-curd), or Laminarin at a concentration of 0.1, 10, 30, 50, 100 or 200 μg/mL per well for 24 h. Two wells per treatment concentration were used, and the assay was repeated two independent times. To determine if the polymers stimulate SEAP release via Dectin-1/Syk/NF-κB signaling, RAW-Blue cells were incubated with 20 μg/ml of either Anti-mDectin-1-IgG antibody (InvivoGen, mabg-mdect) and Rat IgG2a antibody (InvivoGen, mabg2a-ctlrt) for 2 h prior to the addition of Glc-PAS, Gen-PAS-30%, Dextran, Curdlan, lentinan, or Laminarin at 100 μg/mL. To determine if the polymers activity is due to potential contamination with endotoxin, Polymyxin B (10 μg/mL) was added together with the polymers to the cells. After 24 h, 50 μl of the supernatant were added to 150 μL 1X QuantiBlue solution, which was then incubated at 37 °C in the dark for 6 h, including medium only wells to control for potential phosphatase in the media. SEAP concentrations were determined by plate reader at 620–655 nm. Three wells per treatment concentration were used, and the assay was repeated three independent times.

## Assay for macrophage polarization using THP-1 cells

The assay was adapted from the literature with minor changes[90]. THP-1 cells ($1 \times 10^6$ cells) were added into the six well plates per well, and 50 ng/mL of 12-O-tetradecanoylphorbol-l3-acetate (PMA) was added to cell media for 48 h. Activated THP-1 cells were differentiated to either M1 or M2 macrophage by treating with 100 μg/mL of polymer P7 or Laminarin in RPMI media for 24 h. Total RNA in THP-1 cells was extracted using RNeasy Mini Kit (Qiagen 74106), and the total cellular RNA was converted to cDNA by High-Capacity cDNA Reverse Transcription Kit (ThermoFisher Scientific, 4374966). Real-time PCR was performed using diluted cDNA with the Luna® Universal qPCR Master Mix (New England Biolabs Inc, M3003L) and GeneQuery™ Human Macrophage Polarization Markers qPCR Array (ScienCell, Cat#GK120) by the Applied Biosystems StepOnePlus RT-PCR instrument. The expression of each target transcript was calculated using the Delta Delta (ΔΔ) cycle threshold (Ct) method normalized to the expression of the GAPDH housekeeping gene, where ΔΔCt = ΔCt (treated) - ΔCt (control), and the normalized gene expression was calculated by the following equation = $2^{-\Delta\Delta Ct}$. The assay was repeated three independent times.

## Isolation and differentiation of primary human macrophages from PBMC

Whole blood from apheresis was obtained from healthy donors at Boston Children's Hospital's (BCH) Plasma Donation Center. Because donor samples are completely de-identified at BCH prior to handling by research personnel and use in the laboratory, there is no IRB associated with the protocol. PBMCs were isolated via Ficoll gradient (Lymphoprep, StemCell 07801). CD14+ monocytes were then isolated from PBMCs using the MojoSort Human CD14 Selection Kit (Biolegend 480025) according to manufacturer protocol and plated at $5 \times 106$ cells per well in a 6-well plate in RPMI supplemented with 10% (v/v) FBS, penicillin/streptomycin, and 50 ng/ml M-CSF. On days 2, 4, and 6 after plating, culture media was refreshed. On day 6, cells were lifted from

 

the plate using cold PBS and Accutase cell detachment solution (StemCell 07920) and replated at $2.5 \times 10^4$ cells/well in a 96 well plate.

On day 7, cells were given fresh media and treated with either nothing, 100 ng/ml LPS (Sigma–Aldrich, L2654), 20 ng/ml IL-4 (Peprotech, 200-04), or 100 µg/ml PAS polymers or β-glucans. After 24 h of incubation, cells were lifted from the plate with cold PBS and Accutase cell detachment solution and stained for CD68 (StemCell 60105FI) and either CD80 (Biolegend 333611), CD163 (Biolegend 305221), or CD206 (R&D Systems FAB25342P). Cells were then analyzed via flow cytometry (Attune, ThermoFisher Scientific).

## Statistical analysis

The data are presented as means ± SD unless stated otherwise. To determine statistical significance for all experiments, data analysis was performed using a two-tailed unpaired $t$ test (Student's $t$ test) with $P$ values determined using GraphPad Prism software. Values of $p < 0.05$ were considered statistically significant.

## Reporting summary

Further information on research design is available in the Nature Research Reporting Summary linked to this article.

## Data availability

The authors declare that the data supporting the findings of this study are available within the paper and its Supplementary Information file. All other information is available from the corresponding authors upon reasonable request. Correspondence and requests for materials should be addressed to M.W.G.

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

## Acknowledgements

This work was supported in part by Boston University and Boston University Nanotechnology Innovation Center (BUnano), and the National Institutes of Health (NIH RO1EB017722 (M.W.G.), T32GM130546 (E.M.B.), and F30 CA257566 (E.M.B.)).

## Author contributions

R.X. contributed to the polymer design, synthesis, and characterization; J.Z. and E.M.B. contributed to the in vitro experiments and data analysis; W.L. contributed to the GPC characterization of benzylated polymers; R.X., J.Z., E.M.B., and M.W.G. contributed to the experimental design and wrote the paper.

## Competing interests

The authors declare no competing interests.
