## [Peer Review File · Nature Communications]

Synthesis of Bioactive (1-6)- β -Glucose Branched Poly-amido-saccharides that Stimulate and Induce M1 Polarization in Macro-phagesREVIEWER COMMENTS

Reviewer #1 (Remarks to the Author):

Dr. Xiao and colleagues have identified an important problem. Specifically, there is a need for synthetic approaches that can produce high purity, bioactive glucan mimetics with defined structures. The authors have proposed an interesting and potentially innovative approach to solving this problem. However, I find the study to be descriptive in nature. There are also questions about the nature of the mimetics produced in this study. In addition, I do not think the studies go far enough in critically evaluating the working hypothesis. These concerns reduce my overall enthusiasm for this manuscript.

A general concern is the descriptive nature of the study. The reviewer recognizes that studies of this type tend to be descriptive in nature and while this is not a fatal weakness, this manuscript would significantly benefit from mechanistic studies that address some key questions. As an example, the authors claim to have made synthetic compounds which mimic glucan structure and, more importantly, glucan bioactivity. However, the evidence they provide in support of their claim that the PAS compounds mimic glucan bioactivity relies solely on modulation of various cytokines, chemokines and/or macrophage polarization responses. This is not convincing because similar responses can be induced in macrophages by many compounds, both natural and synthetic. What is needed is evidence that the compounds developed during the course of this research mediate their effects via mechanisms that are consistent with a natural product glucan. Specifically, are these glycomimetics recognized and bound by Dectin-1, the primary glucan pattern recognition receptor? Is the bioactivity of these compounds mediated through a Dectin-1/Syk dependent mechanism, etc? If the PAS compounds do not mediate their effects through known glucan mechanism(s)-of-action, then it raises the question of whether these compounds are really glucan mimics.

The Introduction gives the reader the impression that the compounds developed during this research are similar to fungal glucans, as indicated by statements such as “these polymers reflect the composition, branching structure, as well as helical conformation of natural glucans”. This is misleading. What the authors actually made is a poly-amido-saccharide backbone with varying amounts of (1-6)-beta linked side chains, i.e. a dendrimer consisting of peptide and an alpha(?) linked saccharide backbone (scaffold) with (1-6)-beta linked side chains. The structure and composition of their product raises several very interesting questions. The authors state that the (1-6) branches are indispensable for the immunomodulatory effect of the compounds. Does this mean that all (or most) of the bioactivity is attributable to the (1-6) glucan? If so, then what is the role of the poly-amido backbone? It was also noted that the sidechains appear to be one or two subunits in length, which is rather small for induction of bioactivity. All of this is relevant because it has been reported that chemically pure fungal (1-6)-linked glucans (without being attached to a structurally dissimilar backbone) exhibit bioactivity in various in vitro and in vivo models. Thus, one might conclude that the PAS dendrimers are simply a mechanism for delivering and/or presenting (1-6)-glucan to macrophages. This would have been an interesting finding in and of itself, but the authors do not appear to have considered this possibility. Some of these questions could have been answered, at least in part, if the authors had included a (1-6)-glucan control. The authors did include a (1-3,1-6) linked glucan (laminarin) as a positive control. While this is a useful control, it is not the only control that was needed in this study. It should also be noted that (1-6)-beta glucan induced bioactivity does not appear to be mediated through Dectin-1 dependent mechanisms.

The polymers described in this paper are, in all likelihood, not “structurally well defined”. My reason for saying this is based on the authors statements, i.e. “Glu-lactam is more reactive than Gen-lactam.” In the 50/50 ratio Glc-lactam/Gen-lactam polymerization reaction there would not be a β -(1,6)-glucose side-chain every other carbohydrate in the backbone chain. In fact, early in the polymerization, the chain would consist predominantly of poly-Glc-lactam. After most of the Glc-lactam was consumed, more and more of the Gen-lactam would take part in the reaction.

The polydispersity varied considerably between THF and aqueous GPC analysis, i.e. ~11 - 36%. The authors state that this is a minor discrepancy due to under estimation of Mw. A 36% variation is more

than a minor discrepancy, in my opinion. Additional information is required. Employing a light scattering and/or differential viscometry detector in series with the RI during GPC could have resolved some of the issues surrounding M_w and M_n ; and it could also shed additional light on the solution conformation of the polymers.

The issue of the helical nature of the glucan PAS compounds is of interest. The authors are correct that a helical conformation is generally considered to be favorable for induction of glucan induced bioactivity, but this is the subject of ongoing debate. There are numerous reports in the literature of single helical glucans inducing significant bioactivity. The authors provide data suggesting that the glucan PAS compounds may self-anneal and form helical structures. Do all of the polymers form helical structures or only a portion of them? Are the helices stable after manipulation, such as cavitation caused by sterile filtration? Determining the helical conformation of polymers such as glucans in aqueous media is difficult at best. It would be more reassuring if the authors had supporting data, such as Mark Houwink plots.

The doses employed for the in vitro studies are high. Every experiment employed dosages of >50 ug/ml, with some studies using 100 ug/ml or higher. The data indicate that only minimal responses are seen at 50 ug/ml. This is noteworthy because many natural product glucans have been reported to be bioactive at much lower concentrations. The reviewer is aware that some reports have employed similarly high concentrations, however, the relevance of such studies is an open question because it would be almost impossible to achieve such high concentrations in the in vivo tissue milieu. The dose response studies should have been expanded to include dosages of at least 0.1 and 10 ug/ml. If the glycomimetics are not effective at lower dosages then it raises questions regarding bioequivalence of these compounds.

Were the synthetic carbohydrates sterilized and depyrogenated prior to use? Sterilization and removal of residual endotoxin are essential prior to in vitro and/or in vivo testing of carbohydrates. The presence of even low levels of endotoxin, which can be picked during the synthesis via contaminated chemicals, water and/or the ambient environment, can cause significant problems in studies of pro-inflammatory and/or immunomodulatory responses. The compounds should have not only been filter sterilized and depyrogenated, they should have been assayed for sterility and endotoxin levels prior to use.

Reviewer #2 (Remarks to the Author):

This is an interesting manuscript with potentially useful results regarding the immunostimulative properties of these glucans mimetics. The main issue is that the manuscript builds on an introduction about the beta-1,3-glucans and then goes on to design and prepare a polymer that bears little if any resemblance to a beta-1,3-glucan. The polymer is alpha linked, not beta; It is a 1,2-polymer, not 1,3-; and the rigid amide is both longer and conformationally very different to a glycosidic bond. These differences cannot be glossed over and need to be discussed in detail. With the polymeric backbone being so different, it is tempting to conclude that the immunostimulative properties of the beta-1,3-glucans are mainly due to the correct presentation of the branched 1,6-units, but there are enough reports of the activity of unbranched polymers to exclude this. These aspects need to be discussed in greater detail.

Reviewer #3 (Remarks to the Author):

The authors report the synthesis of a new class of β -glucans (β -(1 \rightarrow 6)-glucose branched poly-amido-saccharides (PASs). Although the ring-opening copolymerization to obtain this type of sugars have been reported, the work reported here allows for the efficient synthesis of a novel type of beta-glucan in high purity. The type of beta-glucan demonstrate good to excellent immunomodulatory

activity, as evidence by the enhanced production of tumor necrosis factor (TNF- α) and nitric oxide (NO) and various interleukins (IL-1 β , IL-6, and IL-12A). β -(1 \rightarrow 6)-Glucose-branched PASs are promising synthetic immunomodulators and the studies here may lead to a new type of cancer immunotherapeutic.

The manuscript is well-written, with solid data to support. This reviewer recommends the publication of the manuscript in Nature Communications, with the following suggestions:

1. In the introduction, the authors may want to dig to the depth of the area by presenting specific examples on the biological/medical applications of the beta-glucans, especially, many beta-glucan have been used in clinical studies (e.g. Lentinan, there are many references). The authors may want to discuss what the current results were. Were purity/water solubility the major reason to impede their medicinal applications? What is the purity Lentinan? And what is the minimum dosage of the beta-glucan to be used in biological/clinical studies?
2. There has been a few examples on the synthesis of beta-glucan (e.g. J. Polym. Sci. Part A: Polym. Chem., 2013, 51: 3693-3699 and many others). It is suggested that the authors give/present results from the previous studies, and pointed out how the new method could be better than previous methods.
3. In the introduction, the authors may also want to give a brief introduction on the methods to evaluate the immunological properties of the beta-glucan. Are these methods gold standard to evaluate a polymer's immunological properties?
4. The authors took a lot of words to describe the synthesis and characterization. This part can be largely moved to the Supporting Information.
5. The authors did not mention the control of endotoxin during the course of the experiments, it should noted that endotoxin contamination may give false results.
6. Figure 2, please list the yield in each step (also in Figures 3 and 5), also, the chemical names should be given (e.g. NMI).
7. Figure 4. The GPC curves of polymers P1-P4 in water should be given (also P5-P9). The CD spectra of P1-p4 should be given here as well. These are the polymers that are soluble in water and play the biological functions.
8. Based on the immunological studies, it is certain that the reported polymers are potent immunomodulators. However what is the good value to be clinically meaningful? My question is: are this values good enough to be potentially useful for future biological/animal/clinical studies? This issues should be discussed in the manuscript, especially, Lentinan has been applied clinically and the authors should address the problems/limitations of the current Lentinan.
9. In the summary, it would be better to describe the authors' next plan. Will animal studies be performed? It would be exciting from the animal data that the reported beta-glucans can be potentially developed to novel therapeutics.

REVIEWER COMMENTS

Reviewer #1 (Remarks to the Author):

Dr. Xiao and colleagues have identified an important problem. Specifically, there is a need for synthetic approaches that can produce high purity, bioactive glucan mimetics with defined structures. The authors have proposed an interesting and potentially innovative approach to solving this problem. However, I find the study to be descriptive in nature. There are also questions about the nature of the mimetics produced in this study. In addition, I do not think the studies go far enough in critically evaluating the working hypothesis. These concerns reduce my overall enthusiasm for this manuscript. A general concern is the descriptive nature of the study. The reviewer recognizes that studies of this type tend to be descriptive in nature and while this is not a fatal weakness, this manuscript would significantly benefit from mechanistic studies that address some key questions.

Comment 1: As an example, the authors claim to have made synthetic compounds which mimic glucan structure and, more importantly, glucan bioactivity. However, the evidence they provide in support of their claim that the PAS compounds mimic glucan bioactivity relies solely on modulation of various cytokines, chemokines and/or macrophage polarization responses. This is not convincing because similar responses can be induced in macrophages by many compounds, both natural and synthetic. What is needed is evidence that the compounds developed during the course of this research mediate their effects via mechanisms that are consistent with a natural product glucan. Specifically, are these glycomimetics recognized and bound by Dectin-1, the primary glucan pattern recognition receptor? Is the bioactivity of these compounds mediated through a Dectin-1/Syk dependent mechanism, etc? If the PAS compounds do not mediate their effects through known glucan mechanism(s)-of-action, then it raises the question of whether these compounds are really glucan mimics.

Response 1: We thank the reviewer for the insightful comments. We have performed the additional experiments. Using the RAW-Blue reporter cell line, the PAS possessing 30% (1→6)-β-glucose branches (P7) strongly activates the NF-κB/AP-1 pathway. Meanwhile, blocking of Dectin-1, the major β-glucan receptor, with anti-Dectin-1 antibody results in a significant decrease in the SEAP response elicited by polymer P7 (Figure 6A), indicating that PAS with (1→6)-β-glucose branches are recognized by Dectin-1 and the activation is dependent on Dectin-1 (Figure 6B). These data support the findings that our (1→6)-β-glucose branched PASs are a novel type of β-glucan mimetics.

We have added these results to the revised manuscript on page 5. Please see the Activation of Dectin-1/NF-κB pathway section.

Comment 2: The Introduction gives the reader the impression that the compounds developed during this research are similar to fungal glucans, as indicated by statements such as “these polymers reflect the composition, branching structure, as well as helical conformation of natural glucans”. This is misleading. What the authors actually made is a poly-amido-saccharide backbone with varying amounts of (1-6)-beta linked side chains,

i.e. a dendrimer consisting of peptide and an alpha(?) linked saccharide backbone (scaffold) with (1-6)-beta linked side chains.

Response: We thank the reviewer for the comments. We have edited the sentence to read “these polymers mimic the branching structure of natural β -glucans”.

We do not see our branched PASs as dendrimers, instead they are polymers with a linear pyranose backbone and with (1 \rightarrow 6)- β -glucose branches along the polymer backbone. These polymers share some important structural similarities to (1 \rightarrow 3),(1 \rightarrow 6)- β -glucans, such as lentinan, schizophyllan. The difference is that our PASs are interconnected by (1 \rightarrow 2)- α -amide bond instead of (1 \rightarrow 3)- β -glycosidic bond. As we have discussed in the introduction part (*revised version*), synthesis of β -glucans with (1 \rightarrow 3)- β -glycosidic linkages is very difficult, usually leading to oligomers or stereo-irregular products. Therefore, the traditional strategy to obtain β -glucan mimetics is by introducing (1 \rightarrow 6)- β -sugar branches to linear polysaccharides with various linkages. And previous studies have shown that introduction of β -(1 \rightarrow 6)-sugar branches to linear polysaccharides, such as cellulose and chitin, led to significant increases its antitumor and immunomodulatory activities (*Biomacromolecules* **2011**, *12*, 2267-2274; *Makromolekul Chem* **1985**, *186*, 449-456; *Carbohydr Res* **1992**, *226*, 239-246).

Comment 3: The structure and composition of their product raises several very interesting questions. The authors state that the (1-6) branches are indispensable for the immunomodulatory effect of the compounds. (1) Does this mean that all (or most) of the bioactivity is attributable to the (1-6) glucan?

(2) If so, then what is the role of the poly-amido backbone? It was also noted that the sidechains appear to be one or two subunits in length, which is rather small for induction of bioactivity.

(3) All of this is relevant because it has been reported that chemically pure fungal (1-6)-linked glucans (without being attached to a structurally dissimilar backbone) exhibit bioactivity in various in vitro and in vivo models. Thus, one might conclude that the PAS dendrimers are simply a mechanism for delivering and/or presenting (1-6)-glucan to macrophages. This would have been an interesting finding in and of itself, but the authors do not appear to have considered this possibility. Some of these questions could have been answered, at least in part, if the authors had included a (1-6)-glucan control. The authors did include a (1-3,1-6) linked glucan (laminarin) as a positive control. While this is a useful control, it is not the only control that was needed in this study. It should also be noted that (1-6)-beta glucan induced bioactivity does not appear to be mediated through Dectin-1 dependent mechanisms.

Response: We thank the review for the suggestion. We also changed the word “indispensable” to “important” as shown on page 5.

(1) The results of TNF- α , NO secretion experiments, as well as the new NF- κ B activation show that Glc-PAS, which do not possess β -(1 \rightarrow 6)-branches, very minimally activate macrophages, while the P7 polymer having 30% β -(1 \rightarrow 6)-branches strongly enhances the secretion of TNF- α , NO, as well as NF- κ B (in a Dectin-1 dependent manner). From these results, the presence of the (1 \rightarrow 6)- β -

glucose branches is important for the Dectin-1 recognition and macrophage activation of the PAS polymers. The role of the PAS backbone is likely to present β -(1 \rightarrow 6)-glucose branches to the macrophages.

- (2) Previous studies have shown that the introduction of β -(1 \rightarrow 6)-monosaccharide branches to linear polysaccharides, such as cellulose and chitin, led to significant increases in its antitumor and immunomodulatory activities (*Biomacromolecules* **2010**, *11*, 1212–1216; *Makromol. Chem.* **1985**, *186*, 449–456; *Carbohydr Res* **1992**, *226*, 239–246). A recent study shows that the β -glucans with β -(1 \rightarrow 6)-glucose branches were recognized by Dectin-1 with higher affinity than the comparable linear β -glucans (*J Pharmacol Exp Ther*, **2008**, *325*, 115–23). More recently, Bertozzi *et al* also reported that glycopolypeptides bearing di-glucose units were able to activate RAW264.7 macrophages after conjugation to polystyrene beads (*Angew. Chem. Int. Ed.* **2018**, *57*, 3137–3142). Therefore, sidechains of one or two sugar units in length induce bioactivity.
- (3) In order to explore the macrophage activation pathway and the effect of polymer backbone and branches on immunomodulatory activity, we investigated NF- κ B pathway activation of the PASs, and compared their activity with different glucans with different structures, such as curdlan, laminarin, and dextran.

“We used the RAW-Blue reporter cell line, which are derived from RAW264.7 cells, that stably express secreted alkaline phosphatase (SEAP) upon NF- κ B and activator protein 1 (AP-1) transcriptional activation (*Antimicrob Agents Chemother* **2014**, *58*, 1738–1743). RAW-Blue cells express Dectin-1, the major β -glucan receptor recognizing (1 \rightarrow 3)- β -glucans and (1 \rightarrow 3),(1 \rightarrow 6)- β -glucans. As shown in Figure 6A, incubation of RAW-Blue cells with Dectin-1 agonists, such as curdlan [Cur, (1 \rightarrow 3)- β -glucan] and Lam [(1 \rightarrow 3)(1 \rightarrow 6)- β -glucan], significantly enhance NF- κ B activation, as seen by increased SEAP activity (*Nature*, **2001**, *413*, 36–37; *J Exp Med*, **2002**, *196*, 407–412). On the other hand, dextran (Dex), a (1 \rightarrow 6)- α -glucan, fails to activate the NF- κ B/AP-1 pathways in macrophage, as it is not glucopyranose backbone, and only elicits a slight increase in SEAP expression. This result indicates that the (1 \rightarrow 2)- α -amide bond of Glc-PAS may be not a favorable structural attribute for PRR recognition compared to β -(1 \rightarrow 3)- and β -(1 \rightarrow 6) linkages of natural β -glucans (*J Pharmacol Exp Ther*, **2008**, *325*, 115–123). On the other hand, treatment of RAW-Blue cells with P7 affords a much stronger NF- κ B/AP-1 macrophage activation relative to that of Glc-PAS-exposed cells. Neutralization of Dectin-1 with anti-Dectin-1 antibody significantly decreases the SEAP response elicited by polymer P7 (Figure 6B), indicating that NF- κ B activation depends on the Dectin-1 receptor. Therefore, the presence of the (1 \rightarrow 6)- β -glucose branches is critical for the Dectin-1 recognition and macrophage activation by the PAS polymers, and the (1 \rightarrow 2)- α -amide-linked PAS backbone may act as a scaffold for presenting β -(1 \rightarrow 6)-glucose branches to macrophages. This finding is consistent with previous reports which introduced (1 \rightarrow 6)- β -sugar side chains to cellulose and chitin [β -(1 \rightarrow 4)-glucans] to enhanced antitumor and immunomodulatory activities (*Biomacromolecules*, **2010**, *11*, 1212–1216), and that the presence of β -(1 \rightarrow 6)-glucose branches increases the recognition of β -glucan by Dectin-

1 (*J Pharmacol Exp Ther*, **2008**, 325, 115-123). Addition of polymyxin B (PMB), a LPS inhibitor, does not attenuate the SEAP response elicited by the PAS polymer, further confirming that the macrophage activation is not due to LPS contamination (Figure S10).”

Pustulan, a β -(1 \rightarrow 6)-glucan, was not used as positive control for two reasons: First, branched PASs are an amide-linked β -glucan mimetic with multiple (1 \rightarrow 6)- β -glucose branches of one sugar unit in length, structurally they are more similar to (1 \rightarrow 3),(1 \rightarrow 6)- β -glucan like lentinan, schizophyllan, and laminarin, which have a (1 \rightarrow 3)- β -glucan backbone with multiple (1 \rightarrow 6)- β -glucose branches. In contrast, pustulan is a partially O-acetylated (1 \rightarrow 6)- β -linked linear glucan, and it does not contain multiple (1 \rightarrow 6)- β -glucose terminals. Second, studies have shown that Dectin-1 recognize both (1 \rightarrow 3)- β -glucan and (1 \rightarrow 3),(1 \rightarrow 6)- β -glucan, and β -glucans with β -(1 \rightarrow 6)-glucose branches show higher affinity to Dectin-1 than the comparable linear β -glucan (*J Pharmacol Exp Ther*, 2008, **325**, 115-23). In contrast, a study by Palma *et al* showed that the pustulan does not bind Dectin-1 effectively (*J Biol Chem*. **2006**, 281, 5771-9).

Comment 4: The polymers described in this paper are, in all likelihood, not “structurally well defined”. My reason for saying this is based on the authors statements, i.e. “Glu-lactam is more reactive than Gen-lactam.” In the 50/50 ratio Glc-lactam/Gen-lactam polymerization reaction there would not be a β -(1,6)-glucose side-chain every other carbohydrate in the backbone chain. In fact, early in the polymerization, the chain would consist predominantly of poly-Glc-lactam. After most of the Glc-lactam was consumed, more and more of the Gen-lactam would take part in the reaction.

Response: We thank the reviewer for the comment. In fact, we only used “structurally well-defined” to describe the homopolymer of Gen-lactam. Modification to the context was made to make the description more accurate.

Comment 5: The polydispersity varied considerably between THF and aqueous GPC analysis, i.e. ~11 - 36%. The authors state that this is a minor discrepancy due to under estimation of Mw. A 36% variation is more than a minor discrepancy, in my opinion. Additional information is required. Employing a light scattering and/or differential viscometry detector in series with the RI during GPC could have resolved some of the issues surrounding Mw and Mn; and it could also shed additional light on the solution conformation of the polymers.

Response: Thank you for the comment, and we have performed additional experiments (See SI page 7-9). We measured the molecular weights of P2' and P3' with triple-detection GPC combining refractive index detector (RI), light scattering detector (LS), and viscometer (VISC) (Tosoh EcoSEC Elite Model HLC-8420).

Table S1. Characterization of molecular weight of P2' and P3' with triple detectors (LS, RI and Viscometer)

	Calibration Method	Retention Time ^d	M _n ^e	M _w ^e	M _z ^e	Đ ^f	R _{g,z} ^g	η ^h	R _h ⁱ	R _g /R _h
P2'	LS ^a		46,649	47,466	48,185	1.02	5.6			
	CC ^b	6.668	16,348	17,718	18,691	1.08				1.647
	UC ^c		50,039	50,853	51,643	1.02		0.052	3.4	
P4'	LS		90,396	94,736	97,332	1.05	7.8			
	CC	6.227	29,139	34,502	37,769	1.18				1.472
	UC		91,184	94,965	98,659	1.04		0.099	5.3	

^a light scattering; ^b conventional calibration with RI detector (polystyrene standard); ^c universal calibration with RI detector (polystyrene standard); ^d min; ^e g/mol; ^f M_w/M_n; ^g radii of gyration, nm; ^h intrinsic viscosity (IV), dL/g; ⁱ hydrodynamic radius, nm.

As shown in the Table S1, the MW values obtained with conventional calibration (RI, polystyrene standard) were significantly lower than the MW values obtained with light scattering detector, indicating the molecular weight of benzylated polymers, as measured in Table 1, were underestimated.

Figure S6. Synthesis of Gen-PASs with *tert*-butylacetyl terminal group. Reagents and conditions: (a) *tert*-butylacetyl chloride, LiHMDS, THF, 0 °C, yield: 84%-87%; (b) Na, NH₃ (l), -60 °C, yield: 79%-90%.

In order to further investigate the discrepancy between DP_(GPC) of benzylated polymers and deprotected polymers, we performed the polymerization of Gen-lactam using *tert*-butylacetyl chloride as initiator with [M]/[I] ratios of 15, 25, and 50, and obtained benzylated polymers P14'-P16' and deprotected polymers P14-P16 (Figure S6). *tert*-Butylacetyl chloride was used as initiator because it has nine methyl protons and enables reliable DP_(NMR) analysis by terminal group integration. As we can see from table S3, the DP_(GPC) values of P14'-P16' were significantly lower than the [M]/[I] ratios, but the DP_(NMR)

values of P14'-P16' were in good agreement with the [M]/[I] ratios, indicating underestimation of MW as measured by GPC (RI, polystyrene standard). Again, the DP_(GPC) values of deprotected polymers P14-P16 were higher than the DP_(GPC) values of P14'-P16', which was consistent with the results for P1'-P4' and P1-P4. However, the DP_(NMR) values of P14-P16 were highly consistent with DP_(NMR) values of P14'-P16', indicating that degradation of the polymers did not occur during deprotection. Therefore, the discrepancy between DP_(GPC) values before and after deprotection is due to the different GPC characterization methods.

Table S2. Polymer Characterization using GPC, NMR, and Optical Rotation

Entry	[M]/[I]	M _{n(theo)} ^a	M _{n(GPC)} ^b	DP _(GPC) ^b	Đ ^c	DP _(NMR) ^d	Yield ^e
P14'	15	13479	9700	10.8	1.06	17.2	84
P15'	25	22399	16800	18.7	1.05	28.1	88
P16'	50	44699	35240	39.4	1.05	54.9	87
P14	-	5364	6100	17.1	1.12	18.0	79
P15	-	8874	9600	27.1	1.14	29.7	86
P16	-	17649	17600	49.9	1.23	53.5	90

^a Calculated based on [M]/[I] ratios, g/mol. ^b Determined by THF GPC against polystyrene standards for P14'-P16', or aqueous GPC against dextran standards for P14-P16, g/mol. ^c M_w/M_n. ^d Measured by terminal group integration, °. ^e Isolated yield, %.

Comment 6: The issue of the helical nature of the glucan PAS compounds is of interest. The authors are correct that a helical conformation is generally considered to be favorable for induction of glucan induced bioactivity, but this is the subject of ongoing debate. There are numerous reports in the literature of single helical glucans inducing significant bioactivity.

The authors provide data suggesting that the glucan PAS compounds may self-anneal and form helical structures. Do all of the polymers form helical structures or only a portion of them? Are the helices stable after manipulation, such as cavitation caused by sterile filtration? Determining the helical conformation of polymers such as glucans in aqueous media is difficult at best. It would be more reassuring if the authors had supporting data, such as Mark-Houwink plots.

Response: Indeed, the effect of helical structure (either triple helical or single helical) on the bioactivities of β-glucans is still under debate. While some papers suggest that the secondary conformation of β-glucans does not affect their activities, other studies provided evidence that the helical conformation is an important structural attribute for the immunomodulatory and anti-tumor activities of β-glucans (*Agr Biol Chem*, **1986**, *50*, 2415-2416; *Cancer Res.* **1988**, *48*, 671-5; *Int. J. Orient. Med.*, **1992**, *17*, 57-77;

Immunology Letters **1996**, *52*, 1-7; *Carbohydrate Research* **2000**, *329*, 587–596; *Curr. Med. Chem.*, **2000**, *7*, 715–729; *Carbohydrate Research*, **2005**, *340*, 1515–1521. etc).

Gen-PASs have the same backbone to Glc-PAS ((1→2)- α -amide linked glucose backbone), and our circular dichroism characterization and molecular dynamics (MD) simulations showed that Glc-PAS backbone adopt a left-handed helical conformation in solution. MD simulations also revealed that the amide groups of the Glc-PAS are oriented orthogonal to the helix axis and interact predominantly with solvent water molecules, and the rigid helical conformation of PAS is a consequence of backbone steric constraints resulting from the conformationally restricted pyranose rings (*J. Am. Chem. Soc.* **2016**, *138*, 6532–6540). In this research with the Gen-PAS, we confirmed that this helical conformation is stable: the CD spectrum remains unchanged over a broad pH range from 2.0 to 12.0, or in presence of high ionic strength (2.0 M NaCl), or with protein denaturant (4.0 M urea); increasing the temperature from 5 to 75 °C only leads to a modest decrease in CD intensity. These data are found in Figure 4.

Sterile filtration does not affect the CD spectra of Gen-PASs (Figure S12). The Mark-Houwink plots of deprotected polymers were not obtained, as we were unable to find a water GPC with triple detectors (LS, RI and Viscometry), even though we reached out to several research labs in Boston.

All-atom (AA) model of Glc-PAS 10mer produced using a modified CHARMM force field showed that all the polymer bone folds into helical structure (*J. Am. Chem. Soc.* **2016**, *138*, 6532–6540). However, as shown in Figure 4C of the manuscript, the absolute MRE values of P1 at 187 and 222 nm are slightly smaller than those values for P2–P4, indicating that the terminal residues might be more flexible, as P1 contains a higher proportion of terminal residues (*Angew. Chem., Int. Ed.* **2002**, *41*, 1718).

Comment 7: The doses employed for the in vitro studies are high. Every experiment employed dosages of >50 ug/ml, with some studies using 100 ug/ml or higher. The data indicate that only minimal responses are seen at 50 ug/ml. This is noteworthy because many natural product glucans have been reported to be bioactive at much lower concentrations. The reviewer is aware that some reports have employed similarly high concentrations, however, the relevance of such studies is an open question because it would be almost impossible to achieve such high concentrations in the in vivo tissue milieu. The dose response studies should have been expanded to include dosages of at least 0.1 and 10 ug/ml. If the glycomimetics are not effective at lower dosages then it raises questions regarding bioequivalence of these compounds.

Response: We thank the reviewer for the comments. The immunomodulatory activity of branched PAS (P7) were measured at lower concentration range. And our results show that P7 is able to enhance TNF- α and NO production at concentration as low as 0.1 ug/mL (Figure 5B and 5D).

We agree. There are some papers which used lower beta-glucan concentrations (0-100ug/mL), but there are also many papers using high concentrations (50-1000ug/mL) (*Journal of Functional Foods* **2017**, *37*, 491–500; *International Immunopharmacology* (2008) *8*, 43–50; *Glycoconj J* **2012**, *29*, 365–377; *J. Agric. Food Chem.* **2012**, *60*,

11560–11566; *Int J Biol Macromol* **2012**, 51, 597– 603; *J. Agric. Food Chem.* **2013**, 61, 11400–11409; *Innate Immun.* **2013**, 19, 10–19; *J. Agric. Food Chem.* **2008**, 56, 1442–1450). The goals of our study were to report, for the first time, a novel method to synthesize β -glucan mimetics with tunable FB, Mn, high purity, and high yields, and to describe their immunomodulation activity. These studies provide a rationale for continued development and optimization of these polymers and evaluation of their biological activity.

Comment 8: Were the synthetic carbohydrates sterilized and depyrogenated prior to use? Sterilization and removal of residual endotoxin are essential prior to in vitro and/or in vivo testing of carbohydrates. The presence of even low levels of endotoxin, which can be picked during the synthesis via contaminated chemicals, water and/or the ambient environment, can cause significant problems in studies of pro-inflammatory and/or immunomodulatory responses. The compounds should have not only been filter sterilized and depyrogenated, they should have been assayed for sterility and endotoxin levels prior to use.

Response: All the polymers were filter sterilized before using in bioassays. The endotoxin level was determined and added to the experimental section.

Limulus Amebocyte Lysate Assays. The endotoxin levels of branched PASs (P1-P13), curdlan, laminarin, and dextran polymers were determined by a quantitative end point assay based on the reactivity of gram-negative endotoxin with a modified Limulus Amebocyte Lysate (LAL) and a synthetic color producing substrate to detect endotoxin chromogenically at 37 °C using ToxinSensor™ Chromogenic LAL Endotoxin Assay Kit (Genscript). The standard endotoxin (10 EU/ml) was from E.coli provided in the kit. All the polymers showed endotoxin levels < 0.06 EU/ml.

“Prior to the cell studies, we determined the endotoxin level of the branched PASs (P1-P13), curdlan, laminarin, and dextran polymers using the Limulus Amebocyte Lysate (LAL) assay. All the polymers showed endotoxin levels < 0.06 EU/mL and are considered endotoxin free. “

“Addition of polymyxin B (PMB), a LPS inhibitor, does not attenuate the SEAP response elicited by the PAS polymer, further confirming that the macrophage activation is not due to LPS contamination (Figure S10).”

Reviewer #2 (Remarks to the Author):

Comment 1: This is an interesting manuscript with potentially useful results regarding the immunostimulative properties of these glucans mimetics. The main issue is that the manuscript builds on an introduction about the beta-1,3-glucans and then goes on to design and prepare a polymer that bears little if any resemblance to a beta-1,3-glucan. The polymer is alpha linked, not beta; It is a 1,2-polymer, not 1,3-; and the rigid amide is both longer and conformationally very different to a glycosidic bond. These differences cannot be glossed over and need to be discussed in detail. With the polymeric backbone being so different, it is tempting to conclude that the immunostimulative properties of the beta-1,3-glucans are mainly due to the correct presentation of the branched 1,6-units, but

there are enough reports of the activity of unbranched polymers to exclude this. These aspects need to be discussed in greater detail.

Response: We thank the reviewer for the comments and suggestions. We agree. There are differences in the linkages of the main chain and the PAS backbone may function as a scaffold to present the (1→3)- β -glucose branches to macrophages. In the introduction part, we discussed the structures, properties, as well as the issues with natural β -glucans which limit their study and application, such as difficulty in isolation and purification, low solubility, and batch-to-batch variation. These limitations have motivated the synthesis of β -glucan mimetics. However, the synthesis of polysaccharides with (1→3)- β -linked backbone is challenging. For example, the ring-opening polymerization of 1,3-anhydro-2,4,6-tri-O-(p-bromobenzyl)-O-D-glucopyranose followed by deprotection afforded (1→3)- β -glucan with M_n of only 1200g/mol. Since Dectin-1, the major receptor for β -glucan, recognizes both (1→3)- β -glucan and (1→3),(1→6)- β -glucan, the synthesis of polymers with (1→6)- β -glucose branches has become one of the major method to obtain β -glucan mimetics. Various studies have shown that introduction of β -(1→6)-sugar branches to linear polysaccharides, such as cellulose, chitin, and ivorynut mannan [β -(1→4)-D-mannopyranan] led to significant increases in their antitumor and immunomodulatory activities. Therefore, our branched PASs, which possess a (1→2)- α -amide-linked glucopyranose backbone and (1→6)- β -glucose branches, mimic the branched structures of β -glucans.

In order to further explore the macrophage activation pathway and the effect of polymer backbone and branches on immunomodulatory activity, we performed additional experiments to investigate NF- κ B pathway activation. The PAS, like natural glucans, activates the NF- κ B pathway.

Please see the text on page 6 “We used the RAW-Blue reporter cell line, which are derived from RAW264.7 cells, that stably express in which RAW264.7 cells are transformed to express secreted alkaline phosphatase (SEAP) upon NF- κ B and activator pro-tein 1 (AP-1) transcriptional activation. (*Antimicrob Agents Chemother* **2014**, *58*, 1738-1743). RAW-Blue cells express Dectin-1, the major β -glucan receptor recognizing (1→3)- β -glucans and (1→3),(1→6)- β -glucans. As shown in Figure 6A, incubation of RAW-Blue cells with Dectin-1 agonists, such as curdlan [Cur, (1→3)- β -glucan] and Lam [(1→3)(1→6)- β -glucan], significantly enhance NF- κ B activation, as seen by increased SEAP activity (*Nature*, **2001**, *413*, 36-37; *J Exp Med*, **2002**, *196*, 407-412). On the other hand, dextran (Dex), a (1→6)- α -glucan, fails to activate the NF- κ B/AP-1 pathways in macrophage, as it is not glucopyranose backbone, only elicits a slight increase in SEAP expression. This result indicates that the (1→2)- α -amide bond of Glc-PAS may be not a favorable structural attribute for PRR recognition compared to β -(1→3)- and β -(1→6) linkages of natural β -glucans (*J Pharmacol Exp Ther*, **2008**, *325*, 115-123). On the other hand, treatment of RAW-Blue cells with P7 affords a much stronger NF- κ B/AP-1 macrophage activation relative to that of Glc-PAS-exposed cells. Neutralization of Dectin-1 with anti-Dectin-1 antibody significantly decreases the SEAP response elicited by polymer P7 (Figure 6B), indicating that activation depends on Dectin-1 receptor. Therefore, the presence of the (1→6)- β -glucose branches is critical for the Dectin-1

recognition and macrophage activation by the PAS polymers, and the (1→2)- α -amide-linked PAS backbone may act as a scaffold for presenting β -(1→6)-glucose branches to macrophages. This finding is consistent with previous reports which introduced (1→6)- β -sugar side chains to cellulose and chitin [β -(1→4)-glucans] to enhanced antitumor and immunomodulatory activities (*Biomacromolecules*, **2010**, *11*, 1212-1216), and that the presence of β -(1→6)-glucose branches increases the recognition of β -glucan by Dectin-1 (*J Pharmacol Exp Ther*, **2008**, *325*, 115-123). Addition of polymyxin B (PMB), a LPS inhibitor, does not attenuate the SEAP response elicited by the PAS polymer, further confirming that the macrophage activation is not due to LPS contamination (Figure S10)."

It's true that linear β -glucans like curdlan also exhibit immunomodulatory effect, however, it has also been shown that the introduction of β -(1→6)-sugar branches to linear curdlan led to significant increases its antitumor and immunomodulatory activities (Carbohydrate Research, 1992, 226, 239-246, *Biomacromolecules* 2011, *12*, 2267–2274). A recently study also showed that the β -glucans with β -(1→6)-glucose branches were recognized by Dectin-1 with higher affinity than the comparable linear β -glucan (*J Pharmacol Exp Ther*. 2008, *325*, 115-123.). Therefore, we propose that "the presence of the (1→6)- β -glucose branches is important for the Dectin-1 recognition and macrophage activation of the PAS polymers"

Reviewer #3 (Remarks to the Author):

The authors report the synthesis of a new class of β -glucans (β -(1→6)-glucose branched poly-amido-saccharides (PASs). Although the ring-opening copolymerization to obtain this type of sugars have been reported, the work reported here allows for the efficient synthesis of a novel type of beta-glucan in high purity. The type of beta-glucan demonstrate good to excellent immunomodulatory activity, as evidence by the enhanced production of tumor necrosis factor (TNF- α) and nitric oxide (NO) and various interleukins (IL-1 β , IL-6, and IL-12A). β -(1→6)-Glucose-branched PASs are promising synthetic immunomodulators and the studies here may lead to a new type of cancer immunotherapeutic.

The manuscript is well-written, with solid data to support. This reviewer recommends the publication of the manuscript in Nature Communications, with the following suggestions:
Response: We thank the reviewer for the positive comments, highlighting the immunomodulatory activity of these novel polymers, and recommendation for publication.

Comment 1. In the introduction, the authors may want to dig to the depth of the area by presenting specific examples on the biological/medical applications of the beta-glucans, especially, many beta-glucan have been used in clinical studies (e.g. Lentinan, there are many references). The authors may want to discuss what the current results were. Were purity/water solubility the major reason to impede their medicinal applications? What is the purity Lentinan? And what is the minimum dosage of the beta-glucan to be used in biological/clinical studies?

Response: We thank the reviewer for the comments. We have added additional discussion about clinical studies, the purity and solubility of β -glucans to the introduction.

Please see page 1 “These polysaccharides are not directly cytotoxic to cancer cells, but instead exert tumoricidal effects via activation of the immune system of the host, therefore, possess great therapeutic potentials. For example, schizophyllan and lentinan (Figure 1) are approved in Japan for clinical use in human cancer treatment (*Food Rev Int*, **1995**, *11*, 23-61). Recent clinical studies show that, compared to chemotherapy alone, chemo-immunotherapy with lentinan prolongs the survival of patients with advanced gastric cancer (*World J Clin Oncol*, **2011**, *2*, 339-343).”

“Accumulating evidence demonstrates that the activities of β -glucans are influenced by their frequency of branching (FB), molecular weight (Mn), secondary structure, and solubility, but defining the effect of these structural parameters on biological function is challenging. This is partly due to the use of β -glucans with different structures and from different sources (composition, branching structure and frequency, conformation, and molecular weight), difficulty in structure determination, and presence of impurities (*J Hematol Oncol*, **2009**, *2*, 25).”

“A recent study showed that laminarins purchased from different vendors can be either Dectin-1 agonists or antagonists depending on the physicochemical properties, purity, and structure (*J Immunol*, **2018**, *200*, 788-799).”

“The low solubility of β -glucans is another obstacle for their clinical use, as systemic administration of insoluble or particulate β -glucans can cause significant adverse health effects, such as microembolization, granuloma formation, inflammation and pain, as well as higher sensitivity to endotoxins (*Crit Rev Biotechnol*, **2005**, *25*, 205-230).”

Please see page 7. A previous study showed that lentinan could markedly inhibit the growth of Sarcoma 180 implanted subcutaneous s.c. in mice, inducing almost complete regression of tumors at doses of 1 mg/kg x10 doses with no sign of toxicity (*Cancer Research*, **1970**, *30*, 2776-2781). In a clinical study, 2 mg/body of lentinan was intravenously administered for 30 min every 2 or 3 wk in combination with chemotherapy (S-1) in the chemo-immunotherapy group. Median overall survival was significantly longer in the group receiving chemo-immunotherapy than in the group receiving chemotherapy alone (*World J Clin Oncol*. **2011**, *2*, 339-343).

Comment 2. There has been a few examples on the synthesis of beta-glucan (e.g. J. Polym. Sci. Part A: Polym. Chem., **2013**, *51*: 3693-3699 and many others). It is suggested that the authors give/present results from the previous studies, and pointed out how the new method could be better than previous methods.

Response: We thank the reviewer for the suggestion. The discussion and comparison with conventional polymerization methods to synthesis β -glucan was added to the introduction and conclusion sections.

Please see page 1. “For example, the ring-opening polymerization of 1,3-anhydro-2,4,6-tri-O-(p-bromobenzyl)-O-D-glucopyranose followed by deprotection afforded (1 \rightarrow 3)- β -glucan with Mn of only 1200g/mol. The polycondensation of difunctional 2,3,4-tri-O-acetyl- α -D-glucopyranosyl bromide also resulted in (1 \rightarrow 6)- β -glucan oligomers, although a

recent study showed that microwave irradiation greatly promoted the glycosylation efficiency.”

Please see page 7. “Compared to the conventional polymerization methods to prepare β -glucans, the approach we achieve higher molecular weight, better control over the stereo-regularity, and provide polymers with tunable sugar branches.”

Comment 3. In the introduction, the authors may also want to give a brief introduction on the methods to evaluate the immunological properties of the beta-glucan. Are these methods gold standard to evaluate a polymer’s immunological properties?

Response: We thank the reviewer for the suggestion. Please see page 4. In the introduction to the Immunomodulatory activities section we briefly describe the rationale for measuring TNF- α and NO, and as well as supportive references.

“It is well documented that β -glucans activate the host immune system and raise the functional activities of various innate immune cells.⁴⁷ Macrophages represent the first line in protecting the body against foreign substance and invading pathogens.^{48 49 50} Macrophages express typical cell surface receptors called pattern recognition receptors (PRRs) that recognize the β -glucan component of fungi and bacteria, such as dectin-1, complement receptor 3 (CR3), scavenger receptors (SRs), and Toll-like receptors (TLRs) (*Cell*, **2002**, 111, 927–930). Binding of β -glucans to these PRRs on the surface of macrophages induces the activation of transcription factors such as NF- κ B family, which subsequently mediates elevated expression of inflammatory cytokines and mediators, such as TNF- α , interleukins, and nitric oxide (NO) (*J Biol Chem* **2001**, 276, 20781-20787; *Cell* **2010**, 140, 805-820). Therefore, elevated cytokine production is often used as a key indicator of macrophage activation (*Innate Immun* **2013**, 19, 10-19; *Int Immunopharmacol* **2006**, 6, 317-333). “

Comment 4. The authors took a lot of words to describe the synthesis and characterization. This part can be largely moved to the Supporting Information.

Response: The description of the monomer and polymer synthesis and characterization was shortened as suggested.

Comment 5. The authors did not mention the control of endotoxin during the course of the experiments, it should be noted that endotoxin contamination may give false results.

Response: All the polymers were filter sterilized before using in bioassays. Endotoxin level was assayed and added to the experimental section.

Limulus Amebocyte Lysate Assays. The endotoxin levels of branched PASs (P1-P13), curdlan, laminarin, and dextran polymers were determined by a quantitative end point assay based on the reactivity of gram-negative endotoxin with a modified Limulus Amebocyte Lysate (LAL) and a synthetic color producing substrate to detect endotoxin chromogenically at 37 °C using ToxinSensor™ Chromogenic LAL Endotoxin Assay Kit (Genscript). The standard endotoxin (10 EU/ml) was from E.coli provided in the kit. All the polymers showed endotoxin levels < 0.06 EU/ml.

“We also examined the effect of polymyxin B (PMB), a LPS inhibitor, on the NF- κ B activation elicited by the PAS polymers. As shown in Figure S10, addition of PMB did not attenuate the SEAP response elicited by P7, further confirming that the NF- κ B activation was not due to LPS contamination.”

Comment 6. Figure 2, please list the yield in each step (also in Figures 3 and 5), also, the chemical names should be given (e.g. NMI).

Response: The monomer synthesis scheme was moved to SI (Figure S1), and the yield in each step was listed and chemical names was given and abbreviations were defined.

Comment 7. Figure 4. The GPC curves of polymers P1-P4 in water should be given (also P5-P9). The CD spectra of P1-P4 should be given here as well. These are the polymers that are soluble in water and play the biological functions.

Response: The GPC curves of polymers P1-P4, P5'-P9', as well as the CD spectra of P5-P9 in water were added to Figure 4. The GPC curves of polymers P5-P9, P10'-P13', and P10-P13 were added to SI (Figure S7 and Figure S9).

Comment 8. Based on the immunological studies, it is certain that the reported polymers are potent immunomodulators. However what is the good value to be clinically meaningful? My question is: are this values good enough to be potentially useful for future biological/animal/clinical studies? This issues should be discussed in the manuscript, especially, Lentinan has been applied clinically and the authors should address the problems/limitations of the current Lentinan.

Response: We thank the reviewer for the comment. From the TNF- α and NO secretion experiments, PAS with 30% of glucose branches (P7) exhibits greater immunomodulatory than positive control laminarin. Treatment of RAW-Blue reporter cells also shows P7 is more potent in activating macrophage NF- κ B/AP-1 pathway compared to natural β -glucans laminarin and curdlan. Therefore, P7 is worthy for further animal studies as a synthetic immunomodulatory. Lentinan was not used in current study as a control, because we were unable to obtain pure lentinan product. (The lentinan we purchased from Carbosynth was not pure, and attempts to purify it via dialysis and sephadex chromatography were not successful.)

Although lentinan has been approved for clinical application, recently studies show that clinically used lentinan samples have poor batch-to-batch consistency and contain other impurities, such as proteins and other sugars (*Journal of Pharmaceutical and Biomedical Analysis* **2013**, 78–79,176–182, *International Journal of Biological Macromolecules* **2020**, 159, 129–136).

Lentinan is insoluble in water, and as we discussed in the introduction, systemic administration of insoluble or particulate β -glucans can cause significant adverse health effects such as microembolization, granuloma formation, inflammation and pain, as well as higher sensitivity to endotoxins (*Crit Rev Biotechnol*, **2005**, 25, 205-230). In contrast, our branched PASs are all water-soluble.

Comment 9. In the summary, it would be better to describe the authors' next plan. Will animal studies be performed? It would be exciting from the animal data that the reported beta-glucans can be potentially developed to novel therapeutics.

Response: The antitumor study of branched PASs in treating Sarcoma 180 implanted subcutaneously in mice is planned and we have mentioned this in the revised conclusion text.

REVIEWER COMMENTS

Reviewer #1 (Remarks to the Author):

Dr. Xiao and colleagues have done a great deal of work in responding to my concerns. I commend for their effort. Their responses as well as the new data they have added did answer some of my questions, but not all of them. In addition, some the new data that they have added raised more questions than it answered. I also have concerns about some of the new results and the conclusions drawn based on those results. My specific comments are given below.

1. The authors have attempted to address the mechanism of action of their compounds. The data presented in Fig. 6A suggests that the GP7 compound activates the NFkB/AP-1 pathway in RAW-Blue cells. P7 also appears to be almost twice as potent as laminarin and curdlan at concentrations >100 micrograms/ml. This is impressive, but it is also surprising because curdlan is a potent natural product glucan. It is even more surprising that laminarin and curdlan are bioequivalent in this assay. That is unexpected because curdlan is usually much more potent than laminarins.

2. The antibody neutralization data in Fig. 6B suggests that P7 exerts its effect, in part, through Dectin-1. However, antibody neutralization reduced the P7 response by ~50%. This result could be explained by insufficient antibody blockade of Dectin-1. However, it may indicate that P7 is mediating its effect through receptors and signaling pathways other than Dectin-1. In any case, these data do not strongly support the authors conclusion that "activation (by P7) depends on Dectin-1...".

3. Interestingly, in Fig. 6B anti-Dectin inhibited the laminarin positive control response by approximately 25%, which is very modest. It's not clear why anti-Dectin had such a minimal effect on laminarin bioactivity, but if this result is correct then the laminarin effect is primarily mediated via mechanisms other than Dectin-1. In my opinion, this result is probably due to a methodological problem. Nevertheless, I consider these data to be equivocal. The study would have been more compelling and more definitive if the authors had employed primary monocyte/macrophages or BMDMs from Dectin-1 KO mice.

4. The authors have attempted to address the issue that their mimetics do not truly resemble glucans. Their responses were not particularly convincing. They also point out that the synthesis of glucans is challenging. I cannot speak to this issue, but I did search the literature and found several papers going back more than a decade that describe the de novo synthesis of glucans with the correct structure and stereochemistry. Furthermore, some of these synthetic glucan mimetics had side chains and they were bound by Dectin-1. Based on this new (to me) information it's unclear why the current approach is really necessary and if it is, why is it better than a glucan mimetic which accurately models the natural product?

5. In the initial review, I raised questions about the bioequivalence of the PAS compounds relative to natural product glucans. Specifically, the use of doses 100 micrograms/ml and higher. This question has not been resolved. The data in Fig. 5 clearly show that the synthetic compounds are minimally effective below 100 micrograms/ml. In addition, the TNF α responses above 100 micrograms/ml are modest when compared with many natural product glucans. The authors cite a number of papers that used similarly high doses of various glucans. I conceded that point in the initial review, but it does not mitigate the problem that it would be virtually impossible to achieve such high levels of the compound in vivo due to dilution effects and pharmacokinetic clearance. This is just one of the reasons that glucans have not been successfully translated to the clinical setting. In addition, the authors used laminarin as their glucan positive control, but this may not be the best choice. Commercially available laminarins vary widely in their bioactivity from completely inactive to modest activity. Why not compare the PAS compounds compare to lentinan, schizophyllan or scleroglucan...all of which are water soluble compounds and have been reported to bioactive in vitro and in vivo? Lentinan has also been used clinically.

6. Comment 8. I accept the authors response regarding sterility and endotoxin levels, but it sounds as though the LPS levels were assayed after the fact. In addition, it is stated that curdlan, laminarin and

dextran were also assayed for endotoxin. It is well known that natural product glucans activate the “G factor pathway” in the Limulus assay (see Miyazaki et al J. Clin Lab Anal 9(5):334, 1995). This results in false positives when trying to assay glucans for LPS contamination. In order for most Limulus assays to be non-responsive to glucans, the G factor has to be depleted. It is surprising that the curdlan or laminarin (depending on the Mw) did not produce a false positive.

Reviewer #2 (Remarks to the Author):

My comments have been satisfactorily addressed and i am now happy to recommend publication.

Reviewer #3 (Remarks to the Author):

The authors have nicely addressed my concerns and suggestions, and have revised the manuscript accordingly. I now recommend the publication of the manuscript.

Reviewer comments:**Reviewer #1 (Remarks to the Author):**

Comment. Dr. Xiao and colleagues have done a great deal of work in responding to my concerns. I commend for their effort. Their responses as well as the new data they have added did answer some of my questions, but not all of them. In addition, some the new data that they have added raised more questions than it answered. I also have concerns about some of the new results and the conclusions drawn based on those results. My specific comments are given below.

Response. Thank you for acknowledging our substantial work to respond to the comments. We appreciate your careful and constructive review and input on the manuscript.

Comment 1. The authors have attempted to address the mechanism of action of their compounds. The data presented in Fig. 6A suggests that the GP7 compound activates the NFkB/AP-1 pathway in RAW-Blue cells. P7 also appears to be almost twice as potent as laminarin and curdlan at concentrations >100 micrograms/ml. This is impressive, but it is also surprising because curdlan is a potent natural product glucan. It is even more surprising that laminarin and curdlan are bioequivalent in this assay. That is unexpected because curdlan is usually much more potent than laminarins.

Response 1. We thank the reviewer for the comments. We have repeated the studies, included a positive control, and performed new studies in primary human macrophages. The updated results are shown in Fig. 6A. The results show that laminarin and curdlan were roughly bioequivalent, consistent with previous results. The laminarin also exhibited significantly higher potency than curdlan at 200 µg/ml ($p = 0.015$). We also included a lentinan treatment as a positive control for this assay, and the results show that P7 induced the highest NF-kB activity, consistent with previously observed results. We agree with the reviewer that curdlan is a potent natural product glucan and that the results are surprising. The laminarin we used in these study was purified according to a recent study which found that purification of laminarin samples remarkably increases its biological activities: it could be either a Dectin-1 agonist or antagonist depending on its purity [Williams et al. J Immunol. 2018, 200(2), 788-799].

Further, RAW-Blue cells are neither primary nor human, and NF-kB signalling does not paint a complete picture of cellular polarization as it acts as a generic master regulator of the inflammatory response. As such, we have performed additional analysis using primary human macrophages isolated and differentiated from peripheral blood mononuclear cells (PBMCs) of healthy donors at Boston Children's Hospital, and have examined multiple specific signals related to macrophage polarization (Figure 7, S14, S15, and S16 as well as new text on pages 7 and 8). The results with the primary macrophages are in agreement with the RAW-blue cells.

Comment 2. The antibody neutralization data in Fig. 6B suggests that P7 exerts its effect, in part, through Dectin-1. However, antibody neutralization reduced the P7 response by ~50%. This result could be explained by insufficient antibody blockade of Dectin-1. However, it may indicate that P7 is mediating its effect through receptors and signaling pathways other than Dectin-1. In any case, these data do not strongly support the authors conclusion that "activation (by P7) depends on Dectin-1...".

Response 2. In our previous results, while Anti-Dectin 1 antibody only reduces the response by 50% in raw absorbance, this represents a near complete repression of the response. The assay

produces a basal response that was 50% of the original absorbance of cells treated with P7. We have repeated these results with additional controls and further optimization of assay conditions (Fig. 6B). We increased the Anti-Dectin 1 antibody concentration to 20 $\mu\text{g}/\text{mL}$ (up from 10 $\mu\text{g}/\text{mL}$), lengthened the incubation time to 2 hours prior to the addition of polymers (up from 1 hour), and increased the total assay time to 6 hours (up from 4 hours). These experimental procedure changes allowed for greater antibody coverage, longer equilibration of the antibody, and a more robust response from the RAW-Blue cells. With the adjusted experimental conditions, we showed that Anti-Dectin 1 reduced the NF- κB response of Lentinan, Laminarin, and P7 to basal levels, indicating that these three polymers mediate their effect through Dectin-1 pathway.

In our updated Fig. 6B, we have added a dotted line to represent the average readout of cells that are untreated. All cells, regardless of treatment, exhibit some basal readout, and treatment with Anti-Dectin 1 antibodies represses the response to basal levels for Lentinan, Laminarin and P7.

In agreement with the reviewer's suggestion to change the text, and we have amended the text to say "indicating that activation of NF κB in RAW-Blue cells depends on the Dectin-1 receptor."

Comment 3. Interestingly, in Fig. 6B anti-Dectin inhibited the laminarin positive control response by approximately 25%, which is very modest. It's not clear why anti-Dectin had such a minimal effect on laminarin bioactivity, but if this result is correct then the laminarin effect is primarily mediated via mechanisms other than Dectin-1. In my opinion, this result is probably due to a methodological problem. Nevertheless, I consider these data to be equivocal. The study would have been more compelling and more definitive if the authors had employed primary monocyte/macrophages or BMDMs from Dectin-1 KO mice.

Response 3. We agree with the reviewer that further assay development is needed. As outlined above, the assay has been further optimized which resulted in a more robust effect in assessing Anti-Dectin 1 activity. Further, while the effect was more modest under the original assay conditions, our new data reveals that the basal response of the assay is very similar to the response to P5. With this context, the original assay showed a reduction of ~60% compared to the basal assay readout. Dectin-1 is one of the members of the C-type lectin receptor (CLR) family, which is involved in numerous pathophysiological processes including macrophage polarization and neuroinflammation.

As suggested, we also performed studies using primary human macrophages, which revealed that P7 polarizes macrophages to the M1 phenotype via flow cytometric analysis of CD80, CD163, and CD206 (Figure 7). Specifically, M1-polarized macrophages exhibit significantly higher CD80 expression, moderately lower CD206 expression, and moderately higher CD163 expression. The results from Figure 7 show that P7 exhibited significantly higher CD80 expression, lower CD206 expression, compared to untreated (medium) treated samples and M2-polarized macrophages, and similar CD163 expression compared to M1-polarized macrophages.

Comment 4. The authors have attempted to address the issue that their mimetics do not truly resemble glucans. Their responses were not particularly convincing. They also point out that the synthesis of glucans is challenging. I cannot speak to this issue, but I did search the literature and

found several papers going back more than a decade that describe the de novo synthesis of glucans with the correct structure and stereochemistry. Furthermore, some of these synthetic glucan mimetics had side chains and they were bound by Dectin-1. Based on this new (to me) information it's unclear why the current approach is really necessary and if it is, why is it better than a glucan mimetic which accurately models the natural product?

Response 4. We thank the reviewer for your comments. Some early papers described the de novo synthesis of structurally well-defined β -glucan mimetics and studied the interaction of these materials with Dectin-1 receptor. These β -(1,3)-linked oligosaccharides were recognized by Dectin-1 receptors, but these polymers have not advanced to pre-clinical studies, despite being known for a decade and they have several shortcomings that may limit their application as immunomodulatory agents as discussed below and summarized on page 2 of the revised manuscript along with the added references.

(1) These mimetics are usually synthesized step-by-step, either iteratively or convergently, from monosaccharide starting materials. The syntheses were very challenging and usually involved more than 10 or even 20 step reactions (① Carbohydr. Res. 2009, 344, 439: 6mer, 12 steps; ② Journal of Carbohydrate Chemistry, 2011, 30, 249: 6mer, 12 steps,; ③ Bioorganic & Medicinal Chemistry, 2012, 20, 3898: 16mer, 22 steps; ④ Carbohydrate Research, 2015, 408, 96: 6mer, 15 steps,; ⑤ Bioconjugate Chem. 2015, 26, 466: 12mer, 15 steps; ⑥ Carbohydrate Research. 2019, 482, 107735: 16mer, 27 steps; ⑦ Journal of Carbohydrate Chemistry, 2015, 34, 215: branched trimer, 16 steps). Additionally such multi-step syntheses require months to complete. Due to the multi-step syntheses, these β -(1,3)-linked oligosaccharides were obtained at the milligram or sub-milligram scale in low overall yields (<5%). The GenPAS is synthesized in two steps from the monomer in high yield (>80% for both steps) and a single polymerization reactions affords 100 mg of material for study in a single day.

(2) The length of these previous β -(1,3)-linked oligosaccharides is short (6 to 16 repeat units) and does not enable access to a wide range of polymer lengths to mimic the natural β -(1,3)-glucans. The affinity of the oligosaccharides to Dectin-1 is dependent on the length of the oligosaccharides, and the minimum length required for detectable binding by dectin-1 was an 11-mer (J. Biol. Chem. 2006, 281, 5771). However, even for the longest synthetic β -(1,3)-oligosaccharide (16 and 17mer), their binding activity was 10-fold weaker than that of natural schizophyllan (① Chem. Commun., 2010, 46, 8249; ② Bioorganic & Medicinal Chemistry. 2012, 20, 3898). In another competitive binding study, oligosaccharides with heptasaccharide to decasaccharide lengths inhibited binding of natural β -(1,3)-glucan on the order of mM, while laminarin consisting of ~30 glucose residues and containing only one branch per ~10 glucose residues in the main chain showed inhibitory activity at 22 nM (J. Pharmacol. Exp. Ther., 2008, 325, 115). The GenPAS are prepared by a controlled anionic ring opening polymerization which enables preparation of varying lengths from 10 to 50+.

(3) The synthesis of β -(1,6)-branched β -(1,3)-oligosaccharides is still challenging and not thoroughly investigated given the limited number of reports over decades. The presence of β -(1,6)-glucose branch is very important for the binding to the receptors. For example, decasaccharide with one branched glucose residue showed 100-fold stronger affinity to Dectin-1 than a linear decasaccharide (J. Pharmacol. Exp. Ther., 2008, 325, 115). However, until recently the efforts had been mainly focused on the construction of linear backbone β -(1,3)-chain, and the synthesis of β -(1,6)-branched β -(1,3)-oligosaccharides with length >11, which are more

biologically active, is still rare. This is because the structure of the glycosyl acceptor and donor designed for the introduction of the branched structure increases steric hindrance and disfavors β -(1,3)-bond formation. For example, in the report of Yamamura et al, the glycosylation of β -(1,6)-linked disaccharide acceptor with β -(1,3)-linked disaccharide donor only led to a branched tetrasaccharide in a 15% yield (J. Carbohydr. Chem. 34, 215–246). Enzymatic or chemoenzymatic polymerization of sugar fluoride was also investigated, and could provide linear β -glucan oligosaccharides or polysaccharides with DP up to 30 (① Plant Physiol. 1993, 101, 1131; ② Eur. J. Biochem. 2001, 268, 4628; ③ J. Biol. Chem. 2002, 277, 30102). However, this strategy usually requires expensive sugar fluoride monomers, affords insoluble crystalline or microfibrillar unbranched β -glucan that precipitates from reaction mixture, and could only synthesize β -glucan in small scales. Therefore, progress in this field still needs more efforts in the discovery of more efficient, versatile, and thermostable enzymes and methods to improve activity, scope and yield (Carbohydrate Research. 2021. 508, 108411).

Based on these considerations, our results demonstrate a new polymerization method that provides high molecular weight materials in relatively large scales which are promising β -glucan mimetics. Although the PAS has a different backbone ((1→2)- α -amide linked) compared to natural β -glucan ((1→2)- β -O linked), it still acts as scaffold for presenting the (1→6)- β -glucose branches to Dectin-1 receptor on macrophages. These results are consistent with previous findings that introduction of (1→6)- β -sugar branches to linear polysaccharides with other linkages, such as cellulose ((1→4)- β -O linked), and chitin ((1→4)- β -O linked), increases their antitumor and immunomodulatory activities (① Makromolekul Chem. 1985, 186, 449; ② Carbohydr Res. 1992, 226, 239; ③ Biomacromolecules. 2010, 11, 1212). Meanwhile, compared to the glycosylation strategy, our polymerization method exhibits superior control over the Mn, structure and frequency of branches, and doesn't suffer from degradation and epimerization (Please see page 9 for a short summary of the advantageous of this polymerization method).

Comment 5. In the initial review, I raised questions about the bioequivalence of the PAS compounds relative to natural product glucans. Specifically, the use of doses 100 micrograms/ml and higher. This question has not been resolved. The data in Fig. 5 clearly show that the synthetic compounds are minimally effective below 100 micrograms/ml. In addition, the TNF α responses above 100 micrograms/ml are modest when compared with many natural product glucans. The authors cite a number of papers that used similarly high doses of various glucans. I conceded that point in the initial review, but it does not mitigate the problem that it would be virtually impossible to achieve such high levels of the compound in vivo due to dilution effects and pharmacokinetic clearance. This is just one of the reasons that glucans have not been successfully translated to the clinical setting. In addition, the authors used laminarin as their glucan positive control, but this may not be the best choice. Commercially available laminarins vary widely in their bioactivity from completely inactive to modest activity. Why not compare the PAS compounds compare to lentinan, schizophyllan or scleroglucan...all of which are water soluble compounds and have been reported to bioactive in vitro and in vivo? Lentinan has also been used clinically.

Response 5. We agree that 100 μ g/ml is too high a dose for standard intravenous administration protocols and this is a limitation. We can envision this technology paired with a local drug delivery platform that provides local high doses to increase concentration at the site of action. Our

laboratory and others develop local drug delivery using an implantable device that provides local high doses to increase concentration at the site of action and eliminate the need for intravenous administration (J Control Release 327, 834-856 (2020); *Biomaterials* **219**, 119182 (2019); *Ann Surg Oncol* **17**, 1203-1213 (2010); *Biomaterials* **76**, 273-281 (2016)). Alternatively, one could use a particle delivery format to increase potency and overcome solubility issues. For example, Bertozzi et al conjugated glycopolypeptides with glucose moieties to 0.8 μm polystyrene beads, and found that the conjugates elicited a higher pro-inflammatory response than natural β -glucan curdlan (Angew. Chem. Int. Ed. 2018, 57, 3137–3142). Finally, at the nanoscale, several groups describe β glucan modified nanoparticles for targeting and delivery of active agents (*Int J Nanomedicine* **15**, 5083-5095 (2020); *ACS Omega* **4**, 668-674 (2019); J. Drug Delivery). Future studies will focus on this delivery challenge and we have included new text to highlight this challenge with natural glucans as well as mimetics (see page 9).

Additionally, we have included lentinans as a positive control group for our assays as shown in revised Figures 6 and 7. P7 exhibited a much higher Dectin-1 mediated Nf-kB activity compared to lentinan and also a stronger response in M1 macrophage polarization. Our results support further study and development of (1 \rightarrow 6)- β -glucose branched PASs as immunomodulatory agents. Also, our paper highlights a new polymerization method that enables control the molecular weight, branch structure and frequency, and provides β -glucan mimetics that are free from biological contaminants with batch-to-batch consistency. In fact, this is the first report of polymerization method to synthesize (1 \rightarrow 6)- β -glucose branched polysaccharide mimetics.

Comment 6. Comment 8. I accept the authors response regarding sterility and endotoxin levels, but it sounds as though the LPS levels were assayed after the fact. In addition, it is stated that curdlan, laminarin and dextran were also assayed for endotoxin. It is well known that natural product glucans activate the “G factor pathway” in the Limulus assay (see Miyazaki et al J. Clin Lab Anal 9(5):334, 1995). This results in false positives when trying to assay glucans for LPS contamination. In order for most Limulus assays to be non-responsive to glucans, the G factor has to be depleted. It is surprising that the curdlan or laminarin (depending on the Mw) did not produce a false positive.

Response 5. While β (1 \rightarrow 3) glucans activate LAL endotoxin assays, the effect is glucan type and concentration specific. Zhang et. al. (*Journal of Clinical Microbiology* **32**, 1537-1541 (1994)) found that laminarin concentrations greater than 1 $\mu\text{g}/\text{ml}$ in LAL buffer blocked false positive results from β (1 \rightarrow 3) glucans without sacrificing endotoxin sensitivity, and concentrations of curdlan over 1 mg/ml resulted in a negative result from the LAL assay. Wako/Fujifilm developed an endotoxin-specific (ES) buffer that contains hydroxymethylated curdlan in high concentration to block β (1 \rightarrow 3) glucan activity in the LAL test. Thus, it is not surprising that our initial endotoxin testing was not affected by false positive results, especially since β (1 \rightarrow 3) glucan concentration in these samples was higher than 1 mg/ml.

To ensure our samples were free of endotoxin, we performed the assay on all polymers with and without endotoxin-specific buffer from Wako/Fujifilm. Further, we adjusted the concentration of all polymers to 100 ng/ml to be within the concentration range shown to produce a false positive by Zhang et. al. We saw no interference with the assay while using the endotoxin-specific buffer (Fig. S16A), and we saw that all glucans produced a false positive without the

endotoxin-specific buffer, but they did not produce a false positive with the endotoxin-specific buffer. None of the synthetic polymers produced a true positive response (Fig. S16B).

We have also amended the text under the methods section “Limulus Amebocyte Lysate Assays” to say the following (Page 10): “The endotoxin levels of all polymers and β -glucans used in primary macrophage polarization assays were also examined at concentrations of 100 ng/ml with and without endotoxin specific assay buffer (Wako/Fujifilm, ESB-0006), which contains high concentrations of hydroxymethylated curdlan. Sensitivity of the assay was determined with and without ES buffer (Figure S16A), and all polymers exhibited expected results (Figure S16B).”

A

B

Reviewer #2 (Remarks to the Author):

My comments have been satisfactorily addressed and i am now happy to recommend publication.

Reviewer #3 (Remarks to the Author):

The authors have nicely addressed my concerns and suggestions, and have revised the manuscript accordingly. I now recommend the publication of the manuscript.

REVIEWERS' COMMENTS

Reviewer #1 (Remarks to the Author):

The authors have done an excellent job of responding to and addressing my concerns. Well done!

The response to the reviewer is below.

Reviewer #1

Comment 1: The authors have done an excellent job of responding to and addressing my concerns.
Well done!

Response 1: Thank you.